# GRAPH VISION NETWORKS FOR LINK PREDICTION

## ABSTRACT

The potential of the vision modality for enhancing graph structural awareness has long been overlooked in the mainstream graph neural network (GNN) community. In this paper, we propose a simple yet effective framework called Graph Vision Networks (GVN), which first incorporates vision awareness into Message Passing Neural Network (MPNN) and achieves effective performance for link prediction, highlighting this unexplored but promising direction. Specifically, GVNs transform graph structures into images and extract Visual Structural Features (VSFs) from those images, where VSFs are considered a novel type of structural feature. Similar to previous structural features, VSFs also mitigate the limitations of traditional MPNNs in expressive power and substructure awareness. Additionally, unlike most previous heuristic-based structural features (e.g., common-neighbor-based and path-based ones), which typically depend on fixed structural priors, VSFs are adaptive and capable of capturing varying structural insights to better suit different scenarios. Extensive experiments across seven commonly used benchmark datasets demonstrate that GVNs and their variants can significantly enhance MPNNs in link prediction tasks. Additionally, the straightforward design of the framework makes it highly compatible with current methods, providing additional performance gains to achieve new state-of-the-art performance.

## 1 INTRODUCTION

Link prediction is a fundamental task in graph machine learning, and has been widely used across various application domains. Examples include recommendation systems (He et al., 2020), drug interaction prediction (Yamanishi et al., 2008), and knowledge-based reasoning (Bordes et al., 2013). A class of powerful link predictors are the Graph Neural Networks (GNNs), which produce node representations and then aggregate them to link representations for the prediction of link existence.

While GNNs are very popular, they suffer from **limited expressive power**. In particular, they produce the same representations for links involving isomorphic nodes[1] (Morris et al., 2019; Xu et al., 2018), and ignores the **pairwise structural relations** between the two nodes in the target link (Zhang et al., 2021; Chamberlain et al., 2022; Wang et al., 2024). Second, the **structure awareness** ability of MPNNs is **coarse-grained**. It can be proved that MPNNs are incapable of counting local structural patterns such as triangles (Chen et al., 2020). Empirically, as will be demonstrated in our experiments, MPNNs cannot estimate link prediction heuristics such as Common Neighbor Counts (CN) (Barabási & Albert, 1999), Resource Allocation (RA) (Zhou et al., 2009), and Adamic Adar (AA) (Adamic & Adar, 2003).

To address the aforementioned issues, a number of strategies have been proposed to improve MPNNs for link prediction. One direct approach involves assigning labels or random node features to all nodes, thereby enabling MPNNs to generate distinct node representations for isomorphic nodes and facilitating the differentiation of links involving such nodes. However, this comes at the cost of inductive ability and training convergence (Abboud et al., 2020; Sato et al., 2021; Zhang et al., 2021). A more effective approach involves designing and computing heuristic structural features (HSFs), also known as labeling tricks, that are derived from the local graph structure. These HSFs supplement MPNNs with more detailed and sophisticated structure characteristics, therefore enhancing expressive power and structural awareness. This approach has shown remarkable success on link prediction. For instance, SEAL (Li et al., 2020) utilizes the shortest path distance (SPD) between the nodes (target

---

[1]An example is shown in Appendix A.

nodes) in the target link as HSFs. ID-GNN (You et al., 2021) assigns "identity" colors to target nodes as HSFs. More recently, models such as Neo-GNN (Yun et al., 2021), BUDDY (Chamberlain et al., 2022), and NCNC (Wang et al., 2024) leverage different types of one-hop and multi-hop common neighbor information to construct common-neighbor-based HSFs, leading to state-of-the-art (SOTA) link prediction performance.

Despite the outstanding performance, each HSF is derived from a pre-defined structural prior, thus encapsulating structural information from only **one predefined perspective**. However, real-world situations are **complex and variable**, and may demand structural insights from diverse perspectives. For instance, while SEAL and NBFNet (using path-based HSFs) have superior link prediction performance on the *planetoid* dataset (Yang et al., 2016), they perform even worse than the simple Graph Convolutional Network (GCN) on the *ogbl-ddi* dataset (Hu et al., 2020), which contains dense graphs and most node pairs are reachable in two hops, making path-based HSFs not sufficiently informative. Consequently, due to the fixed structural insights of HSFs, users are often required to try repeatedly to find the best-suited HSF. Hence, there is a growing demand for methods capable of generating **adjustable and adaptive** structural features tailored to different application scenarios. Ideally, this approach should be **compatible** with existing methods that use fixed HSFs, and provide performance enhancements for scenarios with already-identified heuristic preference.

To achieve this, we propose the Graph Vision Network (GVN), which innovatively utilizes the visual modality to extract dynamic and learnable structural features (called *Visual Structural Features*, or VSFs) from the visual representations of graphs, thereby enhancing the expressive power and structural awareness of MPNNs. Specifically, GVN first visualizes local graph structures as visual graph images. A learnable vision encoder is then employed to dynamically extract VSFs from these images. Subsequently, the VSFs are integrated into MPNNs through a learnable attention-based fusion module, which adaptively enhances link prediction for different scenarios. The proposed GVN framework includes two variants: GVN-Link and GVN-Node, where the latter is particularly designed for large graphs. Due to the simple but effective design, both variants are compatible with existing HSF-based methods. We demonstrate that VSF, as a novel type of structural feature, possesses flexible and comprehensive structural awareness. The extensive experiment results on seven common datasets including challenging large-scale graphs demonstrate both GVN-Link and GVN-Node can significantly enhance traditional MPNN in link prediction (28.20% and 36.15% respectively). Besides, when incorporated into existing methods, both GVN-Link and GVN-Node achieve new state-of-the-art (SOTA) performance.

In summary, the contributions of this paper are three-fold.

- We are the first practice to integrate the vision modality into MPNNs for link prediction by proposing a novel structural feature: visual structural features (VSFs), highlighting a promising direction to combine vision awareness into GNNs.
- By incorporating adaptive VSFs to MPNNs, we propose the GVN framework, which has a simple but effective design and is able to be compatible with existing methods.
- Extensive experiments demonstrate that GVNs significantly enhance MPNNs in link prediction and can achieve SOTA performance by further improving existing methods with vision awareness.

## 2  RELATED WORK

**Link Predictor.** Link predictors can be divided into three classes: node embedding methods, link prediction heuristics, and MPNN-based link predictors. *1) Node embedding methods* (Perozzi et al., 2014; Tang et al., 2015; Grover & Leskovec, 2016) represent each node as an embedding vector and utilize the embeddings of target nodes to predict links. *2) Link prediction heuristics* (Liben-Nowell & Kleinberg, 2003; Barabási & Albert, 1999; Adamic & Adar, 2003; Zhou et al., 2009) create structural features through manual design. *3) MPNN-based link predictors* explicitly model the enclosing subgraphs around the nodes through MPNNs and generate/update node embeddings via the message-passing mechanism, thus fully leveraging node attributes and aggregating node representations. However, the expressive power of naive MPNN architectures is proven to be limited (Zhang et al., 2021), constrained by the 1-WL test (Morris et al., 2019), and they fail to finely perceive substructures like triangles (Chen et al., 2020). To overcome these limitations, more advanced MPNNs are proposed that integrate link prediction heuristics and their extended form as structural features (i.e., HSFs) into MPNNs. For instance, SEAL (Zhang & Chen, 2018) incorporates path-based SPD structural

features into MPNNs, which concatenates the SPD from each node to the target nodes $u$ and $v$ with the node features to form the augmented node features $X'$ and apply MPNN on a $k$-hop subgraph $S_{u,v}^k$ centered around $(u, v)$. Other common-neighbors-based HSFs have also been incorporated into MPNNs. For example, Neo-GNN (Yun et al., 2021) and BUDDY (Chamberlain et al., 2022) use the heuristic function to model high-order common neighbor information. NCNC (Wang et al., 2024) directly concatenates the weighted sum of node representations of common neighbors with the Hadamard product of MPNN representations of $u$ and $v$.

**Graph Learning with Vision.** Recently, there are a number of explorations on leveraging vision to enhance graph learning. Das et al. (2023) find that the vision modality, combined with a vision-language model (VLM), can outperform GNN baselines for node classification on the planetoid datasets. Wei et al. (2024) shows that the vision modality excels at capturing graph substructures such as local cycles and triangles with the help of VLMs. However, using vision as structural features with MPNNs or integrating vision in link prediction remains unexplored, which is the focus of this paper.

## 3 PRELIMINARIES

**Notations.** An undirected graph $G = (V, E)$ comprises a set $V$ of $n$ nodes (vertices) and a set $E$ of $e$ links (edges). We denote the adjacency matrix of $G$ by $A \in \mathbb{R}^{n \times n}$, where $A_{(u,v)} > 0$ if and only if the edge $(u, v) \in E$. We define $N(v) := \{v | v \in V, A_{uv} > 0\}$ as the set of neighbors of node $v$, and $N_k(v)$ as the set of neighbors of node $v$ within $k$ hops, where a node $u \in N_k(v)$ if and only if $\text{SPD}(u, v) \leq k$. The node feature matrix $\mathbf{X_G} \in \mathbb{R}^{n \times F}$ contains the node features in $G$, where the $v$-th row $\mathbf{x_v}$ corresponds to the feature of node $v$. We use $S_{uv}^k = (V_{uv}, E_{uv})$ to denote[2] a $k$-hop subgraph enclosing the link $(u, v)$, where $V_{uv}$ is the union of $k$-hop neighbors of $u$ and $v$, and $E_{uv}$ is the union of links that can be reached by a $k$-hop walk originating at $u$ or $v$. Similarly, $S_u^k$ is the $k$-hop subgraph enclosing node $u$.

**Message Passing Neural Networks for Link Prediction.** The MPNN is a common framework for GNNs in link prediction task. In MPNN, the message-passing mechanism is employed to iteratively update node representations based on information exchanged between neighboring nodes. Mathematically, this message-passing mechanism can be written as

$$\boldsymbol{h}_v^t = U^t(\boldsymbol{h}_v^{t-1}, \mathbf{AGG}(\{M^t(\boldsymbol{h}_v^{t-1}, \boldsymbol{h}_u^{t-1}) | u \in N(v)\})), \tag{1}$$

$$\mathbf{Y_G} = \text{MPNN}(\mathbf{X_G}, G), \quad \mathbf{y}_v = \boldsymbol{h}_v^k, \tag{2}$$

where $\mathbf{Y_G} \in \mathbb{R}^{n \times F'}$ is the final node representations by MPNN for graph $G$, whose $v$-th row $\mathbf{y}_v$ is the final representation of node $v$. Given the node representation matrix $\mathbf{Y_G}$, link probabilities can then be computed as $p(u, v) = R(\mathbf{y}_u, \mathbf{y}_v)$, where $R$ is a learnable readout function.

Eqs. (1) and (2) show that MPNNs have *permutation equivariance*, i.e., for any $n \times n$ node permutation matrix $\mathbf{P}$, we have $\mathbf{P}(\text{MPNN}(\mathbf{X}, \text{G})) = \text{MPNN}(\mathbf{PX}, \text{G})$. As a consequence, MPNNs produce the same representation $\mathbf{y}_u = \mathbf{y}_v$ for isomorphic nodes $u$ and $v$. Thus, for any node $w$, $R(\mathbf{y}_w, \mathbf{y}_u) = R(\mathbf{y}_w, \mathbf{y}_v)$, which leads to equal link probabilities $p(w, u) = p(w, v)$ for links $(w, u)$ and $(w, v)$. In other words, MPNNs have limited expressive power.

## 4 GRAPH VISION NETWORKS

In this section, we introduce Graph Vision Networks (GVNs), a novel framework that integrates vision-enhanced MPNNs for link prediction. This includes two variations: GVN-Link and GVN-Node. We provide their framework architecture diagrams in Figure 1

**Problem Setting.** Given an undirected graph $G = (V, E)$ and a set $L$ of query links, the objective of link prediction is to determine the existence of each link $(u, v) \in L$.

### 4.1 GVN-LINK

**Message Passing on Node Features.** GVN-Link initiates the processing pipeline by employing a MPNN to propagate information over the graph $G = \{V, E\}$. This step utilizes the node feature

---

[2]For simplicity, we omit $k$ from $V_{uv}$ and $E_{uv}$.

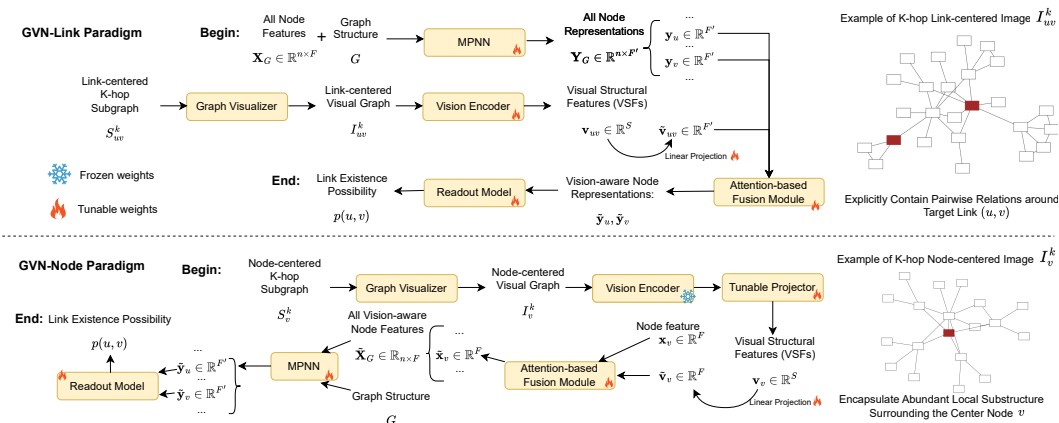

Figure 1: The overview of GVN-Link and GVN-Node architectures

matrix $\mathbf{X_G} \in \mathbb{R}^{n \times F}$, resulting in a node representation matrix $\mathbf{Y_G} \in \mathbb{R}^{n \times F'}$:

$$\mathbf{Y_G} = \text{MPNN}_\phi(\mathbf{X_G}, G),$$

where $\phi$ is the trainable parameter of the MPNN.

**Link-centered Subgraph Visualization.** For each candidate link $(u, v)$, GVN-Link extracts the $k$-hop subgraph $S_{uv}^k$ surrounding it. Subsequently, by using a graph visualizer $GV$ (such as graphviz (Gansner & North, 2000)), the subgraph is transformed to a visual graph image $I_{uv}^k$ with nodes $u$ and $v$ highlighted with special color, as

$$I_{uv}^k = \text{GV}(S_{uv}^k, u, v).$$

An example is shown in the top right of Figure 1. This visual representation encapsulates the structural information around the queried link, such as common neighbors and structural motifs like triangles. These features are expected to be captured by a trained vision encoder in the next step.

**VSF Extraction.** Next, the visual graph image $I_{uv}^k$ is processed through a trainable vision encoder, denoted $\text{VE}_\psi$, to extract the visual structural features $\mathbf{v}_{uv} \in \mathbb{R}^S$:

$$\mathbf{v}_{uv} = \text{VE}_\psi(I_{uv}^k).$$

**Feature Integration.** The extracted VSFs $\mathbf{v}_{uv}$ are then integrated with the node representations $\mathbf{y}_u$ and $\mathbf{y}_v$ by an attention-based fusion module $\text{FM}_\omega$. The following describes the feature integration procedure for node $u$. Processing for node $v$ is similar.

The feature integration process begins with projecting $\mathbf{v}_{uv} \in \mathbb{R}^S$ to $\tilde{\mathbf{v}}_{uv} \in \mathbb{R}^{F'}$ using a linear projector layer, ensuring that $\tilde{\mathbf{v}}_{uv}$ shares the same dimensions as the node representations $\mathbf{y}_u$ and $\mathbf{y}_v$:

$$\tilde{\mathbf{v}}_{uv} = \text{Projector}(\mathbf{v}_{uv}).$$

Subsequently, an attention mechanism evaluates the relevance of the visual features by facilitating selective emphasis on significant visual details and computing the attention vector $\mathbf{y}_u^{attn}$:

$$\mathbf{y}_u^{attn} = \text{attention}(\mathbf{Q} = \mathbf{y}_u, \mathbf{K} = \tilde{\mathbf{v}}_{uv}, \mathbf{V} = \tilde{\mathbf{v}}_{uv}) = \text{softmax}\left(\frac{\mathbf{y}_u \tilde{\mathbf{v}}_{uv}^T}{\sqrt{F'}}\right) \tilde{\mathbf{v}}_{uv}.$$

The integration of VSFs with node representations is refined through a weighted combination, regulated by a learnable parameter $\alpha$ that balances the original and visually enhanced features:

$$\tilde{\mathbf{y}}_u = \alpha \mathbf{y}_u + (1 - \alpha)\mathbf{y}_u^{attn}.$$

The feature integration procedure for node $u$ can be summarized as $\tilde{\mathbf{y}}_u = \text{FM}_\omega(\mathbf{y}_u, \mathbf{v}_{uv})$, where the trainable parameters $\omega$ in the fusion module $\text{FM}$ include the parameters of the linear projector layer and attention layer, as well as the scaling parameter $\alpha$.

**Link Probabilities Read-out.** The read-out model $R_\theta$, which integrates the enhanced node representations $\tilde{\mathbf{y}}_u$ and $\tilde{\mathbf{y}}_v$, computes the probability of the existence of a link $(u, v)$:

$$p_{(u,v)} = R_\theta(\tilde{\mathbf{y}}_u, \tilde{\mathbf{y}}_v).$$

## 4.2 GVN-NODE

**Node-centered Subgraph Visualization.** Contrary to GVN-Link, GVN-Node employs the graph visualizer GV to visualize the node-centered $k$-hop subgraph $S_v^k$ for each node $v$. An example is shown in the bottom right of Figure 1.

$$I_v^k = \text{GV}(S_v^k, v).$$

**Partially-trained VSF Extraction.** For each node $v$, the visual image of its node-centered subgraph is subsequently converted into the node-based VSF $\mathbf{v}_v \in \mathbb{R}^S$ by the vision encoder, which reflects the local structural features surrounding the node. In GVN-Node, a "partial training strategy" is employed for VSF extraction, which aims to save computational cost while keeping the VSFs adaptive. To be specific, we make the parameters of the vision encoder fixed, but add a trainable linear projector appended to the vision encoder to make the VSFs still trainable. As a result, we can store the intermediate $\text{VE}(I_v^k)$ as a vector database, to save the time of frequently loading and processing the images by VE per epoch.

$$\mathbf{v}_v = \text{Projector}_\psi(VE(I_v^k)).$$

**Feature Integration.** For each node $v$, the node-based VSF $\mathbf{v}_v$ is then integrated into its original node feature $\mathbf{x}_v$ through the same attention-based fusion module $\text{FM}_\omega$ used in GVN-Link. The integration updates $\mathbf{x}_v \in \mathbb{R}^F$ to a vision-aware node feature $\tilde{\mathbf{x}}_\mathbf{v} \in \mathbb{R}^F$:

$$\tilde{\mathbf{x}}_v = \text{FM}_\omega(\mathbf{x}_v, \mathbf{v}_v) = \alpha \mathbf{x}_v + (1 - \alpha)\mathbf{x}_v^{attn}, \quad \tilde{\mathbf{v}}_v = \text{Projector}(\mathbf{v}_v),$$

where

$$\mathbf{x}_v^{attn} = \text{attention}(\mathbf{Q} = \mathbf{x}_v, \mathbf{K} = \tilde{\mathbf{v}}_v, \mathbf{V} = \tilde{\mathbf{v}}_v) = \text{softmax}\left(\frac{\mathbf{x}_v \tilde{\mathbf{v}}_v^T}{\sqrt{F}}\right)\tilde{\mathbf{v}}_v.$$

**Message Passing on Vision-aware Node Features.** Subsequently, the MPNN is used to perform message passing on graph $G$ with these vision-aware node features and output the final node representations, where $\tilde{\mathbf{X}}_\mathbf{G}$ is the matrix form of vision-aware node features for all nodes:

$$\tilde{\mathbf{Y}}_\mathbf{G} = \text{MPNN}_\phi(\tilde{\mathbf{X}}_\mathbf{G}, G).$$

**Link Probabilities Read-out.** Finally, the learnable read-out model $R_\theta$ predict the link existence probability with the vision-aware node representations $\tilde{\mathbf{y}}_u$ and $\tilde{\mathbf{y}}_v$:

$$p_{(u,v)} = R_\theta(\tilde{\mathbf{y}}_u, \tilde{\mathbf{y}}_v).$$

## 4.3 TIME COMPLEXITY ANALYSIS

Let $n$ be the number of nodes, $d$ be the maximum node degree, $F$ be the node feature dimension, $F'$ be the dimension of node representation produced by MPNN, and $l$ be the number of target links. The time complexity of GVN-Link is determined by the following components: 1) Complexity of the base model, which includes the MPNN and its associated read-out function. For example, the complexity of GCN is $O(ndF + nF^2) + O(lF^2)$. For the NCNC model (Wang et al., 2024) that incorporates common-neighbor HSFs, the complexity is $O(ndF + nF^2) + O(ld^2F + ldF^2)$. We denote this part by $O(\text{Base})$. 2) Complexity of generating visual images for the target links is $O(l)$. 3) Complexity of extracting visual structural features with Vision Encoder is $O(l)$. 4) Complexity of linear projection which converts the $S$ dimensional VSFs $\mathbf{v}_{uv}$ to the $F$ dimensional $\tilde{\mathbf{v}}_{uv}$ is $O(lSF)$ 4) Complexity of the attention mechanism is $O(lF^2)$. Therefore, the total time complexity of GVN-Link is $O(\text{Base}) + O(l) + O(lSF) + O(lF^2) = O(\text{Base}) + O(lSF + lF^2)$.

For GVN-Node, the difference lies in its use of node-centered subgraph VSFs. Therefore, the complexity of generating visual images for all nodes is $O(n)$, the complexity of the VSF projection becomes $O(nSF')$, the complexity of the attention mechanism becomes $O(nF'^2)$, and the other parts remain the same with GVN-Link. As a result, the total time complexity of GVN-Node is $O(\text{Base}) + O(n) + O(nSF') + O(nF'^2) = O(\text{Base}) + O(nSF' + nF'^2)$.

## 4.4 Comparison between GVN-Link and GVN-Node

GVN-Link and GVN-Node have their own advantages and disadvantages. First, in most graphs, the number of links $l$ is significantly larger than the number of nodes $n$ (with $l$ having an upper bound of $n^2$). Consequently, GVN-Node demonstrates a higher computational efficiency compared to GVN-Link, making it more suitable for large and dense graphs, where GVN-Link can become computationally intensive. Second, the VSFs in GVN-Link include the visual perception of the target link's neighborhood structure. These VSFs explicitly reveal the pairwise relationship between the two nodes in the target link, which is advantageous for link prediction. Third, the VSFs in GVN-Node encompass the visual perception of all nodes' neighborhoods and participate in message passing. This allows the VSFs in GVN-Node to capture more structural details and integrate them more deeply into the link prediction process. In contrast, GVN-Link only provides visual perception of the substructures surrounding the two nodes in the target link.

It is worth noting that although the VSFs in GVN-Node do not explicitly model pairwise relationships, the base model can still learn these relationships from the neighborhood connections between nodes, such as through similar substructures, as a compensation. Furthermore, employing an MPNN method with pairwise HSFs, such as NCNC, as the base model can effectively address this limitation.

## 5 Experiments

In this section, we conduct a series of comprehensive and engaging experiments to demonstrate the effectiveness of the proposed GVN and VSFs.

### 5.1 Evaluation on Real-World Datasets

In this section, we comprehensively evaluate both GVN-Link and GVN-Node with different base MPNN models on seven widely-used datasets, comparing them with a representative set of baselines.

**Datasets.** We conduct experiments on widely used Planetoid citation networks: *Cora* (McCallum et al., 2000), *Citeseer* (Sen et al., 2008), and *Pubmed* (Namata et al., 2012), and the OGB link prediction datasets (Hu et al., 2020): *ogbl-collab*, *ogbl-ppa*, *ogbl-citation2* and *ogbl-ddi*. Statistics of those datasets are shown in Appendix B.

**Baselines.** Baseline methods used include three popular link prediction heuristics: Common Neighbor counts (CN) (Barabási & Albert, 1999), Adamic-Adar (AA) (Adamic & Adar, 2003), and Resource Allocation (RA) (Zhou et al., 2009); two popular GNNs: GraphSAGE (Hamilton et al., 2017) and Graph Convolutional Network (GCN) (Kipf & Welling, 2016); HSF-enhanced GNNs: SEAL (Zhang & Chen, 2018) and NBFNet (Zhu et al., 2021) (which are MPNNs with path-based HSFs), Neo-GNN (Yun et al., 2021), BUDDY (Chamberlain et al., 2022), and NCNC (Wang et al., 2024) (which are enhanced by common-neighbor-based HSFs).

**Configurations of Proposed Methods.** We study four configurations of the proposed GVN-Link (denoted by GVN-L) and GVN-Node (denoted by GVN-N): GVN-L$_{GCN}$, GVN-L$_{NCNC}$, GVN-N$_{GCN}$, and GVN-N$_{NCNC}$, where the subscript denotes the base model (i.e., GCN or NCNC). Note that the proposed methods can be easily applied to other MPNN models. Graphviz (Gansner & North, 2000) (with details in Appendix C) is used as the graph visualizer. We use a pretrained ResNet50 (He et al., 2016) as the vision encoder and extract visual features from its last convolutional layer.

**Performance Evaluation.** The use of evaluation metrics follows (Chamberlain et al., 2022; Wang et al., 2024). Specifically, for the Planetoid datasets, we use the hit-ratio at 100 (HR@100), while for the OGB datasets, we use the metrics in their official documents [3], i.e., hit-ratio at 50 (HR@50) for *ogbl-collab*, HR@100 for *ogbl-ppa*, Mean Reciprocal Rank (MRR) for *ogbl-citation2* and hit-ratio at 20 (HR@20) for *ogbl-ddi*. [4] All results are averaged over 10 trials with different random seeds. Experiments are conducted on an NVIDIA A100 80G GPU. More details on the experimental setup are in Appendix E.

---

[3]https://ogb.stanford.edu/docs/leader_linkprop/

[4]Evaluations on other metrics are included in Appendix D

Table 1: Link prediction performance (average score ± standard deviation). "-" indicates that the training time is > 24 hour/epoch (for GVN-L) or out of memory (for NBFnet). The best performance is shown in bold, and the second-best is underlined.

| | Cora (HR@100) | Citeseer (HR@100) | Pubmed (HR@100) | Collab (HR@50) | PPA (HR@100) | Citation2 (MRR) | DDI (HR@20) |
|---|---|---|---|---|---|---|---|
| CN | $33.92_{\pm0.46}$ | $29.79_{\pm0.90}$ | $23.13_{\pm0.15}$ | $56.44_{\pm0.00}$ | $27.65_{\pm0.00}$ | $51.47_{\pm0.00}$ | $17.73_{\pm0.00}$ |
| AA | $39.85_{\pm1.34}$ | $35.19_{\pm1.33}$ | $27.38_{\pm0.11}$ | $64.35_{\pm0.00}$ | $32.45_{\pm0.00}$ | $51.89_{\pm0.00}$ | $18.61_{\pm0.00}$ |
| RA | $41.07_{\pm0.48}$ | $33.56_{\pm0.17}$ | $27.03_{\pm0.35}$ | $64.00_{\pm0.00}$ | $49.33_{\pm0.00}$ | $51.98_{\pm0.00}$ | $27.60_{\pm0.00}$ |
| SAGE | $55.02_{\pm4.03}$ | $57.01_{\pm3.74}$ | $39.66_{\pm0.72}$ | $48.10_{\pm0.81}$ | $16.55_{\pm2.40}$ | $82.60_{\pm0.36}$ | $53.90_{\pm4.74}$ |
| GCN | $66.79_{\pm1.65}$ | $67.08_{\pm2.94}$ | $53.02_{\pm1.39}$ | $44.75_{\pm1.07}$ | $18.67_{\pm1.32}$ | $84.74_{\pm0.21}$ | $37.07_{\pm5.07}$ |
| GVN-L$_{GCN}$ | $81.13_{\pm0.86}$ | $83.93_{\pm0.97}$ | $73.17_{\pm1.02}$ | - | - | - | - |
| GVN-N$_{GCN}$ | $80.01_{\pm1.55}$ | $82.85_{\pm1.90}$ | $71.94_{\pm1.37}$ | $62.14_{\pm1.37}$ | $32.15_{\pm1.58}$ | $86.10_{\pm0.13}$ | $60.21_{\pm6.67}$ |
| Neo-GNN | $80.42_{\pm1.31}$ | $84.67_{\pm2.16}$ | $73.93_{\pm1.19}$ | $57.52_{\pm0.37}$ | $49.13_{\pm0.60}$ | $87.26_{\pm0.84}$ | $63.57_{\pm3.52}$ |
| SEAL | $81.71_{\pm1.30}$ | $83.89_{\pm2.15}$ | $75.54_{\pm1.32}$ | $64.74_{\pm0.43}$ | $48.80_{\pm3.16}$ | $87.67_{\pm0.32}$ | $30.56_{\pm3.86}$ |
| NBFnet | $71.65_{\pm2.27}$ | $74.07_{\pm1.75}$ | $58.73_{\pm1.99}$ | - | - | - | $4.00_{\pm0.58}$ |
| BUDDY | $88.00_{\pm0.44}$ | $92.93_{\pm0.27}$ | $74.10_{\pm0.78}$ | $65.94_{\pm0.58}$ | $49.85_{\pm0.20}$ | $87.56_{\pm0.11}$ | $78.51_{\pm1.36}$ |
| NCNC | $89.65_{\pm1.36}$ | $93.47_{\pm0.95}$ | $81.29_{\pm0.95}$ | $\underline{66.61}_{\pm0.71}$ | $\underline{61.42}_{\pm0.73}$ | $89.12_{\pm0.40}$ | $\underline{84.11}_{\pm3.67}$ |
| GVN-L$_{NCNC}$ | $90.70_{\pm0.56}$ | $94.12_{\pm0.58}$ | $82.17_{\pm0.77}$ | - | - | - | - |
| GVN-N$_{NCNC}$ | $\mathbf{91.47}_{\pm0.36}$ | $\mathbf{94.44}_{\pm0.53}$ | $\mathbf{84.02}_{\pm0.55}$ | $\mathbf{68.14}_{\pm0.75}$ | $\mathbf{63.45}_{\pm0.66}$ | $\mathbf{90.72}_{\pm0.24}$ | $\mathbf{87.31}_{\pm3.04}$ |

**Implementation Details.** The adjustable key hyperparameters include the vision-aware hop count $k$ ranging from 1 to 3, the hidden dimension ranging from 512 to 2048, the number of MPNN layers and readout predictor layers varying from 1 to 3, the separate two learning rates for learning trainable VSFs and adaptive fusion model among 0.0000001, 0.00001, 0.001, 0.01 and the weight decay from 0 to 0.0001. The hyperparameters with the best validation accuracy are selected. For the model parameters, we utilize the Adam optimizer (Kingma, 2014) to optimize them. All results of our models are derived from runs using 10 different random seeds.

**Results.** Table 1 compares the performance of the proposed methods with the various baselines. As can be seen, integration of VSF through either GVN-Link or GVN-Node consistently enhances link prediction performance across both base models. In particular, with GCN as the base model, GVN-L$_{GCN}$ and GVN-N$_{GCN}$ boost the performance dramatically relative to the GCN baseline (with an average improvement of 28.20% for GVN-L$_{GCN}$ on the Planetoid datasets, and 36.15% for GVN-N$_{GCN}$ on all seven benchmarks). This remarkable enhancement underscores the value of VSFs as dynamic structural features that significantly boost the capabilities of MPNNs. On the other hand, when NCNC is used as the base model, GVN-L$_{NCNC}$ and GVN-N$_{NCNC}$ achieve new SOTA performance, illustrating that VSFs can provide additional enhancements that are compatible with existing SOTA methods.

With GCN as the base model, GVN-Link outperforms GVN-Node. This is mainly because the link-centered VSFs in GVN-Link are more adept at elucidating the pairwise structural relationships surrounding these links compared to the node-centered VSFs in GVN-Node. Conversely, with NCNC as the base model, GVN-Node outperforms GVN-Link. This is because NCNC's HSFs compensate for the lack of explicit pairwise information in node-centered VSFs, allowing GVN-Node to utilize its refined structural perception capabilities effectively. GVN-Node's VSFs focus on node-level structural details and are iteratively refined through message passing, resulting in a more detailed understanding of the local structures compared to the link-centered VSFs in GVN-Link. As a result, GVN-N$_{NCNC}$ outperforms GVN-L$_{NCNC}$.

In terms of applicability, GVN-Link is best suited for smaller graphs (such as *Cora*, *CiteSeer*, and *PubMed*) due to its expensive runtime (exceeding 24 hours per epoch on the larger graphs). Conversely, GVN-Node demonstrates better scalability and broader applicability, showing effectiveness even on large-scale graph datasets.

## 5.2 DELVE INTO THE REASON WHY INCORPORATING VISION HELPS LINK PREDICTION

We delve into the reasons how vision helps MPNN-based link prediction from two aspects:

**Vision alleviates two limitations of MPNNs in link prediction.** 1) Distinguish Links with Isomorphic Nodes. As highlighted in Section 1, MPNNs exhibit limited expressive power due to their

inability to differentiate links with isomorphic nodes (see the illustration in Appendix A), thus producing the same link prediction $p(u, v)$ for these links that have distinct enclosing subgraphs. To illustrate the improvement of GVNs over MPNNs in this regard, we compute the proportion of links that produce the same prediction (i.e., $p(u, v)$) with at least one other link by GVN-L$_{GCN}$, GVN-N$_{GCN}$, and GCN during link prediction on the non-attributed[5] *Cora* dataset. We expect to find whether there are cases where links with distinct enclosing subgraphs are treated the same by GCN but corrected by VSFs in GVN. Empirically, **6.79%** links in GCN share identical predictions with the other links. In contrast, the ratio is only **0.88%** (**0.94%**) for GVN-L$_{GCN}$ (respectively, for GVN-N$_{GCN}$). Therefore, the gap between the ratios demonstrates that GVNs make the links with distinct enclosing subgraphs distinguishable by incorporating VSFs.

2) Capture the substructures. Besides, MPNNs are proven to have only coarse-grained structural awareness on substructures like triangles. To evaluate whether GVNs achieve progress in this aspect, we extract 3200 triangles from the *Cora* dataset and then randomly sample another 3200 non-triangle triplets as negative samples. Each model is then tasked with distinguishing triangles from the negative samples. Empirically, GVN-L$_{GCN}$ achieves an accuracy of **91.88%**, and GVN-N$_{GCN}$ achieves an accuracy of **88.91%**, while GCN attains only **63.25%**. This underscores the more fine-grained structural perception of GVN models compared to traditional MPNNs.

**VSFs encapsulate diverse structural information and can be tailored to specific scenarios.**

Unlike traditional heuristic structural features (HSFs) that depend on a single structural prior like common-neighbors (e.g., CN, RA, AA) or path information between target nodes (e.g., SPD), VSFs offer a rich array of structural insights from multiple perspectives, as illustrated in Figure 2. The adaptability of VSFs allows them to shift focus based on varying scenarios. To validate this adaptability, we explore whether VSFs can be fine-tuned to better suit the current scenario. Specifically, for each link $(u, v)$, we extract link-centered VSFs $\mathbf{v}_{uv}$ in GVN-Link (GVN-L$_{GCN}$) and node-centered VSFs $\mathbf{v}_u$ and

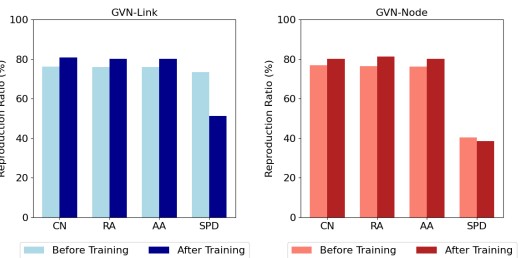

Figure 2: Heuristic Reproduction Ratio (%) Using VSFs in GVN-Link and GVN-Node Before and After Finetuning on *Cora* for Link Prediction.

$\mathbf{v}_v$ in GVN-Node (GVN-N$_{GCN}$), both before and after fine-tuning on the *Cora* dataset. We then evaluate the extent to which these VSFs can replicate link prediction heuristics such as CN, RA, AA, and SPD.

For GVN-L$_{GCN}$, we employ a trainable 3-layer MLP predictor that uses link-centered VSFs as input to predict heuristics. Successful replication of a heuristic $s_{uv}$, i.e., $\text{MLP}(\mathbf{v}_{uv}) = s_{uv}$, indicates that the relevant heuristic information is embedded within the link-centered VSFs $\mathbf{v}_{uv}$. In the case of GVN-N$_{GCN}$, an additional GCN is used for message passing through node-centered VSFs before applying the MLP predictor, i.e., $\text{MLP}(\text{GCN}(\mathbf{v}_u), \text{GCN}(\mathbf{v}_v)) = s_{uv}$, since node-centered VSFs participate in message passing within GVN-N$_{GCN}$.

Figure 2 displays the proportions of heuristics that VSFs can reproduce before and after fine-tuning on the *Cora* link prediction scenario. The results reveal how the type of information contained in VSFs evolves through fine-tuning. Post fine-tuning, VSFs in both GVN-L$_{GCN}$ and GVN-N$_{GCN}$ demonstrate an improved ability to capture common-neighbor-based heuristics (CN, RA, and AA), while their capacity to replicate the path-based heuristic SPD decreases. This suggests that the VSFs learned from the *Cora* scenario prioritize common neighbor information over shortest path information.

This trend aligns with existing observations (Zhang & Chen, 2018; Yun et al., 2021; Chamberlain et al., 2022; Wang et al., 2024) that common-neighbor-based methods (such as BUDDY and NCNC) often outperform SPD-based methods like SEAL for link prediction on the *Cora* dataset. These findings suggest that GVN can dynamically adjust the information in their VSFs to provide more relevant insights tailored to the specific scenario.

---

[5]Here we delete node attributes to make models only focus on the graph structures because here we only care about their expressive power on structure.

## 5.3 SCALABILITY

Figure 3 compares the time and GPU memory for inferring one batch of samples from *Cora* (the preprocessing time for each method is also taken into account).

Among the baselines and proposed methods, GVN-Link (GVN-L$_{GCN}$ and GVN-L$_{NCNC}$) is the most time-consuming, followed by SEAL and NBFnet. These three methods also require considerably more memory than the others. This elevated resource consumption is due to the need for pre-processing and computation for each link, and the storage of intermediate variables with respect to links. Additionally, GVN-Link requires graph visualization, which introduces extra pre-processing time. Therefore, similar to SEAL and NBFnet, GVN-Link is not well-suited for large-scale graph computations.

In contrast, although GVN-Node also requires graph visualization, it is a node-based method which involves fewer computations than link-based methods and allows reuse across different links in the entire dataset. In Figure 3, we include the amortized time for graph visualization in the time cost computation of GVN-Node. As a result, GVN-Node (GVN-N$_{GCN}$ and GVN-N$_{NCNC}$) still exhibits computational overhead **similar to their base models** (GCN and NCNC). Therefore, by leveraging the lightweight base models GCN and NCNC, GVN-Node maintains efficiency and is suitable for large-scale graphs.

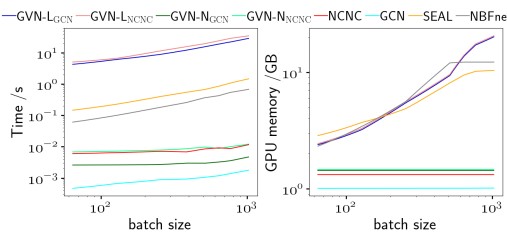

Figure 3: Inference time and memory on Cora.

## 5.4 ABLATION AND SENSITIVITY ANALYSIS

Table 2: Performance comparison on the number of vision-aware hops $k$ (HR@100).

| | *Cora* | | | *Citeseer* | | |
| --- | --- | --- | --- | --- | --- | --- |
| | $k = 1$ | $k = 2$ | $k = 3$ | $k = 1$ | $k = 2$ | $k = 3$ |
| GVN-L$_{NCNC}$ | 89.76±0.78 | **90.70**±**0.56** | 89.68±0.97 | 92.51±0.86 | **94.12**±**0.58** | 92.33±0.91 |
| GVN-N$_{NCNC}$ | 90.87±0.47 | **91.47**±**0.36** | 90.21±0.58 | 93.29±0.59 | **94.44**±**0.53** | 92.62±0.67 |

Table 3: HR@100 on different vision encoder fine-tuning strategies.

| | *Cora* | | | *Citeseer* | | |
| --- | --- | --- | --- | --- | --- | --- |
| | w/o | partial | full | w/o | partial | full |
| **GVN-L**$_{NCNC}$ | 89.57±0.62 | 90.52±0.65 | **90.70**±**0.56** | 91.55±0.79 | 93.75±0.75 | **94.12**±**0.58** |
| **GVN-N**$_{NCNC}$ | 89.66±0.54 | 91.47±0.36 | **91.53**±**0.55** | 92.72±0.48 | 94.44±0.53 | **94.52**±**0.67** |

Table 4: HR@100 performance with different fusion strategies.

| | *Cora* | | | *Citeseer* | | |
| --- | --- | --- | --- | --- | --- | --- |
| | attention | concat | MoE | attention | concat | MoE |
| **GVN-L**$_{NCNC}$ | **90.70**±**0.56** | 85.65±6.25 | 89.99±1.64 | **94.12**±**0.58** | 86.33±4.18 | 93.93±1.04 |
| **GVN-N**$_{NCNC}$ | **91.47**±**0.36** | 76.58±7.79 | 90.66±1.56 | **94.44**±**0.53** | 88.25±5.54 | 93.88±1.21 |

**Sensitivity Analysis of Vision-Aware Hop Count $k$.** Prior research suggests that up to 3 hops usually capture valuable information (Zeng et al., 2021) for MPNNs, where hops are akin to the numbers of graph convolutional layers. However, in our proposed GVNs, the hop of the link-centered or node-centered subgraph, i.e., the hop count that VSFs can be aware of in the vision modality, is decoupled from the MPNN layer counts. Therefore, it is necessary to re-explore how the number of hops that VSFs can be aware of in the vision modality (vision-aware hop count $k$) influences link prediction performance. Table 2 shows the effects of vision-aware hop count $k$ on GVN-L$_{NCNC}$ and GVN-N$_{NCNC}$ on *Cora* and *Citeseer*. Results indicate peak performance for both at $k = 2$, suggesting that a hop count of 2 suffices for VSFs, which aligns with findings for MPNNs.

**Fine-tuning Strategies for Vision Encoder.** Table 3 compares different fine-tuning strategies for the vision encoder: "w/o" uses the pretrained encoder directly, "partial" fine-tunes only a linear projector, and "full" fine-tunes all the encoder parameters. The results illustrate that not fine-tuning ("w/o")

Table 5: HR@100 performance with different vision encoders.

|  | Cora | | | Citeseer | | |
| --- | --- | --- | --- | --- | --- | --- |
|  | ResNet50 | VGG | ViT | ResNet50 | VGG | ViT |
| GVN-L$_{NCNC}$ | **90.70**±0.56 | 89.99±1.61 | 90.69±0.44 | 94.12±0.58 | 93.93±1.35 | **94.29**±1.07 |
| GVN-N$_{NCNC}$ | **91.47**±0.36 | 89.92±1.01 | 91.24±0.66 | 94.44±0.53 | 93.96±0.85 | **94.52**±0.97 |

Table 6: HR@100 performance with different node labeling schemes.

|  | Cora | | | Citeseer | | |
| --- | --- | --- | --- | --- | --- | --- |
|  | No-label | Re-label | Unique | No-label | Re-label | Unique |
| GVN-L$_{NCNC}$ | **90.70**±0.56 | 89.86±0.44 | 89.67±0.62 | **94.12**±0.58 | 94.01±0.43 | 94.08±0.99 |
| GVN-N$_{NCNC}$ | **91.47**±0.36 | 89.73±0.24 | 89.75±0.83 | **94.44**±0.53 | 93.85±0.65 | 94.02±0.91 |

leads to performance drops for GVN-L$_{NCNC}$ and GVN-N$_{NCNC}$, highlighting VSF learnability's significance. Besides, fully fine-tuning achieves the best overall performance.

However, for GVN-N$_{NCNC}$, fully fine-tuning is not always practical. That is because fully fine-tuning requires storing all the images in memory, which can cause out-of-memory issues with larger graphs. For instance, full fine-tuning of GVN-Node on *ogbl-citation2* requires an additional 410.47GB for storing all images. An alternative approach to achieve full fine-tuning on these larger datasets is to dynamically load images into memory in batches, but this increases the time to over 24 hours per epoch for the OGB datasets. To balance efficiency and performance, given that partial fine-tuning for GVN-Node can achieve nearly the same effectiveness as full fine-tuning but is much more efficient in terms of both time and memory, we advocate for partial fine-tuning with GVN-Node.

**Effect of Fusion Strategies.** In this experiment, we study the effectiveness of different fusion strategies, including (i) attention, (ii) concatenation, and (iii) Mixture of Experts (MoE) (Jacobs et al., 1991). Implementation details are in Appendix F.1. Table 4 shows the HR@100 performance of GVN-L$_{NCNC}$ and GVN-N$_{NCNC}$ with these three different fusion strategies on *Cora* and *Citeseer*. Concatenation shows much inferior performance and higher standard deviation than attention. This could be attributed to the trivial handling of the unaligned embeddings of VSFs and graph features. Similarly, MoE is also worse than attention, indicating that jointly managing VSFs and node features using attention is more effective than treating them as separate experts.

**Effect of Vision Encoders.** In this experiment, we study the robustness of the proposed methods with the choice of vision encoder. Three popular encoders are used: (i) ResNet50 (as used in previous experiments), (ii) VGG16 (Simonyan & Zisserman, 2014), and (iii) ViT (Dosovitskiy et al., 2021). Table 5 shows the HR@100 performance of GVN-L$_{NCNC}$ and GVN-N$_{NCNC}$ with these three vision encoders on *Cora* and *Citeseer*. As can be seen, the choice of vision encoder may slightly affect the performance, but does not influence the effectiveness of VSFs and GVN with all of them still outperforming baselines.

**Node Labels in Graph Visualization.** In this experiment, we study different ways to label the nodes in the image: (i) "No-label", which shows the nodes without any labels; (ii) "Re-label", which maps all the nodes in the current subgraph to new labels starting from zero; (iii) "Unique", which labels the nodes with unique global indices. Example images for these visualization schemes are shown in Appendix F.2. Table 6 shows the HR@100 performance of GVN-L$_{NCNC}$ and GVN-N$_{NCNC}$ with these different labeling schemes on *Cora* and *Citeseer*. As can be seen, "no-label" performs best, indicating that purely using the structural information is preferred.

## 6 CONCLUSION

We propose the Graph Vision Networks (GVN) framework, which innovatively incorporates vision features as a new type of structural feature, termed visual structural features (VSFs), to enhance MPNNs in link prediction tasks. Unlike previous methods that rely on fixed heuristic structural priors, VSFs are adaptively extracted and fused to suit the current scenario and are also compatible with existing methodologies. Experimental results demonstrate that VSFs are both informative and adaptive, leading to significant performance improvements beyond base models. Building on the previous SOTA model NCNC, both GVN-Link and GVN-Node achieve SOTA performance. In our future work, we are interested in extending GVN to other graph tasks.

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

## A AN EXAMPLE ILLUSTRATING MPNN'S LIMITED EXPRESSIVE POWER

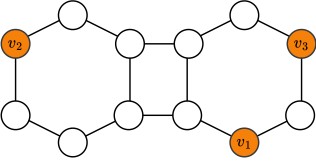

In Figure 4, nodes $v_2$ and $v_3$ are isomorphic because of their symmetric positions in the graph, and they have the **same** $h$-hop neighborhoods for any $h$. Hence, without node features, permutation-equivariant MPNNs produce the same node representations for $v_2$ and $v_3$ (i.e., $\mathbf{y}_{v_2} = \mathbf{y}_{v_3}$). As a result, when predicting distinct links $(v_1, v_2)$ and $(v_1, v_3)$, the input fed into the Readout function are exactly the same (i.e., $(\mathbf{y}_{v_1}, \mathbf{y}_{v_2}) = (\mathbf{y}_{v_1}, \mathbf{y}_{v_3})$). Therefore, the same predictions are produced for the two links.

Figure 4: Example graph with isomorphic nodes.

However, the two links can have **distinct** pairwise structural relations w.r.t. the target node ($v_1$ in this case). For example, $v_3$ is closer to $v_1$ than $v_2$. This difference in structural relations is overlooked by an MPNN but can be effectively captured by the SPD structural feature ($SPD(v_1, v_2) = 5$, $SPD(v_1, v_3) = 2$). Similarly, common-neighbor-based heuristics such as CN, RA and AA can also help the MPNN to distinguish the two links $(v_1, v_2)$ and $(v_1, v_3)$, as $v_1$ and $v_2$ share no common neighbor while $v_1$ and $v_3$ have one.

Therefore, structural features can enhance the expressive power of MPNN in link prediction, by providing extra structural information which are ignored by MPNNs.

## B DATASET STATISTICS

The statistics of the datasets are shown in Table 7.

Table 7: Statistics of dataset.

|  | **Cora** | **Citeseer** | **Pubmed** | **Collab** | **PPA** | **DDI** | **Citation2** |
|---|---|---|---|---|---|---|---|
| #Nodes | 2,708 | 3,327 | 18,717 | 235,868 | 576,289 | 4,267 | 2,927,963 |
| #Edges | 5,278 | 4,676 | 44,327 | 1,285,465 | 30,326,273 | 1,334,889 | 30,561,187 |
| data set splits | random | random | random | fixed | fixed | fixed | fixed |
| average degree | 3.9 | 2.74 | 4.5 | 5.45 | 52.62 | 312.84 | 10.44 |

## C GRAPH VISUALIZER

Graphviz (Gansner & North, 2000) is a powerful tool used for creating visual representations of abstract graphs and networks. It allows for the customization of the styles of nodes, edges, and various layouts (with different configurations of predefined layout computation algorithms, called layout engines) to tailor the visualization to specific requirements.

In our implementation, the graph visualization processes for both GVN-Link and GVN-Node are developed using Graphviz. Besides the typical workflow of using Graphviz to generate graph images (refer to the official documentation at `https://graphviz.org/documentation/`), there are several key configurations in our implementations to prevent their variability from affecting performance robustness:

- **Layout:** For all experiments, we adopt the fixed layout engine "sfdp". This fixed setting makes the results more reproducible and reduces the gap between the training and testing sets. However, we leave a usable interface for specifying other layout engines if needed.

- **Node:** For all experiments, we use a fixed rectangular (box) style for nodes, leaving them empty except for the two nodes in the target link, which are filled with a brown color.

- **Edge:** We treat all edges as undirected links and leave the edge thickness at the default setting. For link-centered subgraph visualization in GVN-Link, the target link is masked.

These configurations help standardize the visualization process, ensuring consistency and clarity in the graphical representations used throughout our work.

## D    EVALUATION ON OTHER METRICS

We test our model in different metrics. The results are shown in Table 8. In total, GVN-Node achieves 39 best scores (in bold), GVN-Link achieves 7 best scores, and our strongest baseline NCNC achieves 3 best scores. Therefore, our GVN-L$_{NCNC}$ and GVN-N$_{NCNC}$ still significantly outperform baselines in different metrics.

Table 8: Models' performance with various metrics. NCNC is our strongest baseline.

| | | Cora | Citeseer | Pubmed | Collab | PPA | Citation2 | DDI |
|---|---|---|---|---|---|---|---|---|
| hit@1 | GVN-L$_{NCNC}$ | $\mathbf{11.75}_{\pm\mathbf{7.72}}$ | $51.69_{\pm7.91}$ | $\mathbf{18.66}_{\pm\mathbf{8.85}}$ | - | - | - | - |
| | GVN-N$_{NCNC}$ | $8.66_{\pm4.39}$ | $\mathbf{59.30}_{\pm\mathbf{5.53}}$ | $16.88_{\pm9.58}$ | $\mathbf{11.04}_{\pm\mathbf{3.01}}$ | $6.53_{\pm1.52}$ | $\mathbf{86.62}_{\pm\mathbf{1.04}}$ | $\mathbf{0.42}_{\pm\mathbf{0.08}}$ |
| | NCNC | $10.90_{\pm11.40}$ | $32.45_{\pm17.01}$ | $8.57_{\pm6.76}$ | $9.82_{\pm2.49}$ | $\mathbf{7.78}_{\pm\mathbf{0.36}}$ | $84.66_{\pm1.15}$ | $0.16_{\pm0.07}$ |
| hit@3 | GVN-L$_{NCNC}$ | $26.66_{\pm5.96}$ | $59.97_{\pm6.21}$ | $\mathbf{32.23}_{\pm\mathbf{5.69}}$ | - | - | - | - |
| | GVN-N$_{NCNC}$ | $\mathbf{27.55}_{\pm\mathbf{6.37}}$ | $\mathbf{66.76}_{\pm\mathbf{4.20}}$ | $31.21_{\pm5.98}$ | $\mathbf{26.31}_{\pm\mathbf{7.74}}$ | $\mathbf{18.88}_{\pm\mathbf{1.21}}$ | $\mathbf{94.29}_{\pm\mathbf{0.96}}$ | $\mathbf{2.12}_{\pm\mathbf{0.33}}$ |
| | NCNC | $25.04_{\pm11.40}$ | $50.49_{\pm12.01}$ | $17.58_{\pm6.57}$ | $21.07_{\pm5.46}$ | $16.58_{\pm0.60}$ | $92.37_{\pm0.56}$ | $0.59_{\pm0.42}$ |
| hit@10 | GVN-L$_{NCNC}$ | $\mathbf{58.83}_{\pm\mathbf{5.29}}$ | $75.28_{\pm3.03}$ | $40.34_{\pm2.28}$ | - | - | - | - |
| | GVN-N$_{NCNC}$ | $55.98_{\pm4.14}$ | $\mathbf{77.12}_{\pm\mathbf{2.95}}$ | $\mathbf{47.90}_{\pm\mathbf{2.86}}$ | $43.12_{\pm5.77}$ | $\mathbf{31.16}_{\pm\mathbf{1.67}}$ | $\mathbf{97.07}_{\pm\mathbf{1.01}}$ | $\mathbf{50.88}_{\pm\mathbf{11.35}}$ |
| | NCNC | $53.78_{\pm7.33}$ | $69.59_{\pm4.48}$ | $34.29_{\pm4.43}$ | $\mathbf{43.22}_{\pm\mathbf{6.19}}$ | $26.67_{\pm1.51}$ | $96.99_{\pm0.64}$ | $45.64_{\pm14.12}$ |
| hit@20 | GVN-L$_{NCNC}$ | $\mathbf{70.01}_{\pm\mathbf{4.44}}$ | $81.11_{\pm1.30}$ | $53.33_{\pm2.67}$ | - | - | - | - |
| | GVN-N$_{NCNC}$ | $69.55_{\pm3.46}$ | $\mathbf{82.02}_{\pm\mathbf{1.46}}$ | $\mathbf{56.92}_{\pm\mathbf{2.33}}$ | $56.87_{\pm2.97}$ | $\mathbf{44.06}_{\pm\mathbf{2.03}}$ | $\mathbf{98.17}_{\pm\mathbf{0.97}}$ | $\mathbf{87.31}_{\pm\mathbf{3.04}}$ |
| | NCNC | $67.10_{\pm2.96}$ | $79.05_{\pm2.68}$ | $51.42_{\pm3.81}$ | $\mathbf{57.83}_{\pm\mathbf{3.14}}$ | $35.00_{\pm2.22}$ | $97.22_{\pm0.94}$ | $83.92_{\pm3.25}$ |
| hit@50 | GVN-L$_{NCNC}$ | $82.06_{\pm1.94}$ | $88.88_{\pm0.98}$ | $\mathbf{71.66}_{\pm\mathbf{2.75}}$ | - | - | - | - |
| | GVN-N$_{NCNC}$ | $\mathbf{82.99}_{\pm\mathbf{2.95}}$ | $\mathbf{88.97}_{\pm\mathbf{0.58}}$ | $71.55_{\pm1.19}$ | $\mathbf{68.14}_{\pm\mathbf{0.75}}$ | $\mathbf{52.58}_{\pm\mathbf{0.30}}$ | $\mathbf{99.09}_{\pm\mathbf{0.66}}$ | $\mathbf{95.95}_{\pm\mathbf{0.75}}$ |
| | NCNC | $81.36_{\pm1.86}$ | $88.60_{\pm1.51}$ | $69.25_{\pm2.87}$ | $66.88_{\pm0.66}$ | $48.66_{\pm0.18}$ | $99.01_{\pm0.53}$ | $94.85_{\pm0.56}$ |
| hit@100 | GVN-L$_{NCNC}$ | $90.70_{\pm0.56}$ | $94.12_{\pm0.58}$ | $82.17_{\pm0.77}$ | - | - | - | - |
| | GVN-N$_{NCNC}$ | $\mathbf{91.47}_{\pm\mathbf{0.36}}$ | $\mathbf{94.44}_{\pm\mathbf{0.53}}$ | $\mathbf{84.02}_{\pm\mathbf{0.55}}$ | $70.83\pm2.25$ | $\mathbf{63.45}_{\pm\mathbf{0.66}}$ | $\mathbf{99.51}_{\pm\mathbf{0.39}}$ | $\mathbf{97.99}_{\pm\mathbf{0.27}}$ |
| | NCNC | $89.05_{\pm1.24}$ | $93.13_{\pm1.13}$ | $81.18_{\pm1.24}$ | $\mathbf{71.96}_{\pm\mathbf{0.14}}$ | $62.02_{\pm0.74}$ | $99.37_{\pm0.27}$ | $97.60_{\pm0.22}$ |
| mrr | GVN-L$_{NCNC}$ | $\mathbf{24.66}_{\pm\mathbf{4.51}}$ | $62.74_{\pm6.63}$ | $26.32_{\pm6.67}$ | - | - | - | - |
| | GVN-N$_{NCNC}$ | $23.27_{\pm3.39}$ | $\mathbf{66.49}_{\pm\mathbf{3.53}}$ | $\mathbf{27.11}_{\pm\mathbf{5.88}}$ | $\mathbf{18.04}_{\pm\mathbf{3.01}}$ | $\mathbf{19.66}_{\pm\mathbf{0.11}}$ | $\mathbf{90.72}_{\pm\mathbf{0.24}}$ | $\mathbf{13.32}_{\pm\mathbf{2.75}}$ |
| | NCNC | $23.55_{\pm9.67}$ | $45.64_{\pm11.78}$ | $15.63_{\pm4.13}$ | $17.68_{\pm2.70}$ | $14.37_{\pm0.06}$ | $89.12_{\pm0.40}$ | $8.61_{\pm1.37}$ |

## E    EXPERIMENTAL SETUPS

**Link Prediction Setups**

In link prediction, links play dual roles: serving as supervision and acting as message-passing paths. Following the standard practice in link prediction, training links fulfill both supervision labels and message-passing paths. In terms of supervision, the training, validation, and testing links are mutually exclusive. For message passing, we follow the common setting where the validation links in *ogbl-collab* additionally function as message-passing paths during test time.

For the Planetoid datasets (*Cora*, *Citeseer*, and *Pubmed*), since the official data splits are not available, we adopt the common random splits of 70%/10%/20% for training/validation/testing. For the OGB benchmarks *ogbl-collab*, *ogbl-ppa*, *ogbl-ddi*, and *ogbl-citation2* (Hu et al., 2020), we utilize the official fixed splits.

For the baselines, we directly use the results reported in (Wang et al., 2024) since we adopt the same experimental setup.

## F    SUPPLEMENTARY DETAILS IN ABLATION STUDY

### F.1    IMPLEMENTATIONS OF FUSION STRATEGIES

**"concat" and "mixture of experts" in GVN-Link.** In GVN-Link, feature integration occurs after message passing and the different fusion strategies will affect how to integrate vision structural awareness to the output of MPNNs.

In GVN-Link, the "concat" strategy is achieved by concatenating the link-centered VSF with the target node features directly, and then use the concatenated features as input to the Readout function. Thus, for each node pair $(u, v)$ in the target link, their MPNN node representations ($\mathbf{y}_u$ and $\mathbf{y}_v$) are

updated to $\tilde{\mathbf{y}}_u$ and $\tilde{\mathbf{y}}_v$ as:

$$\tilde{\mathbf{y}}_u = \mathbf{y}_u || \mathbf{v}_{uv}, \quad \tilde{\mathbf{y}}_v = \mathbf{y}_v || \mathbf{v}_{uv},$$

where $\mathbf{y}_{uv}$ is the link-centered VSF.

For the Mixture of Experts (MoE) fusion strategy, we use a three-layer multilayer perceptron (MLP) as vision-based expert for link prediction. This MLP relies solely on the vision modality by only receiving VSFs as input to output the link prediction probability $p_{(u,v)}^{\text{vision}}$. The MPNN produces another link prediction probability $p_{(u,v)}^{\text{graph}}$ based on message passing and node attributes. Finally, the mixture-predicted link probability $p_{(u,v)}$ is computed as a weighted sum of $p_{(u,v)}^{\text{vision}}$ and $p_{(u,v)}^{\text{graph}}$, combining the capabilities of both the vision-based expert and the MPNN expert with a learnable weight balance parameter $\delta$:

$$p_{(u,v)} = \delta \cdot p_{(u,v)}^{\text{vision}} + (1 - \delta) \cdot p_{(u,v)}^{\text{graph}}.$$

**"concat" and "mixture of experts" in GVN-Node** In GVN-Node, feature integration occurs on the node feature before MPNN, and different fusion strategies use different approaches to obtain the vision-aware node feature $\tilde{\mathbf{x}}_v$.

The "concat" fusion strategy obtains the vision-aware node feature of node $v$ by appending the node-centered VSF $\mathbf{v}_v$ after the central node's features $\mathbf{x}_v$.

$$\tilde{\mathbf{x}}_v = \mathbf{x}_v || \mathbf{v}_v.$$

For "mixture of experts", we use two linear experts to encode the original node feature $\mathbf{x}_v$ and the corresponding node-centered VSF $\mathbf{v}_v$. Therefore, computation of the vision-aware node feature can be expressed as:

$$\tilde{\mathbf{x}}_v = \delta \text{Linear}_{\phi_1}(\mathbf{x}_v) + (1 - \delta)\text{Linear}_{\phi_2}(\mathbf{v}_v),$$

where $\delta$ is a learnable parameter to balance the contributions from the two linear experts, $\text{Linear}_1$ and $\text{Linear}_2$ are linear experts with trainable parameters $\phi_1$ and $\phi_2$.

## F.2 ILLUSTRATIONS FOR DIFFERENT LABELING SCHEMES

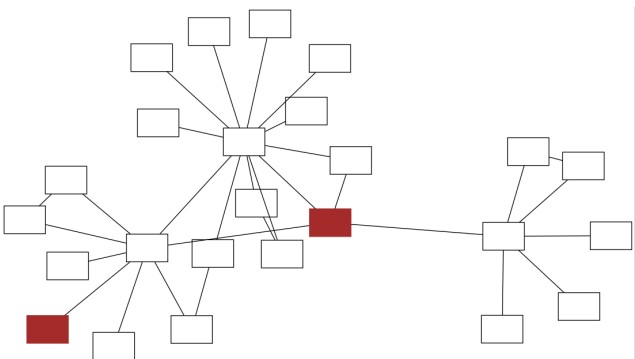

Figure 5: Link-centered subgraph visualization with "No-label" labeling scheme.

In this Section, we present image examples for graph visualization using both GVN-Link and GVN-Node with various labeling schemes.

Figures 5-7 show an example of link-centered subgraph visualization in GVN-Link with various labeling schemes, where the target link is $(1, 158)$. This indicates the objective is to predict the existence of a link between node 1 and node 158. Similarly, Figures 8-10 present node-centered subgraph visualization images with various labeling schemes, where the colored node is the center node.

In Figures 5 and 8, the "No-label" labeling scheme is applied. In this scheme, node labels are omitted, enabling the model to focus purely on the intrinsic graph topological structural information, which is beneficial for generalizability across different datasets or settings.

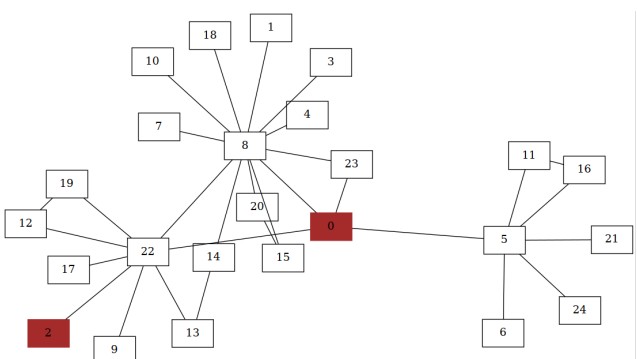

Figure 6: Link-centered subgraph visualization with "Re-label" labeling scheme.

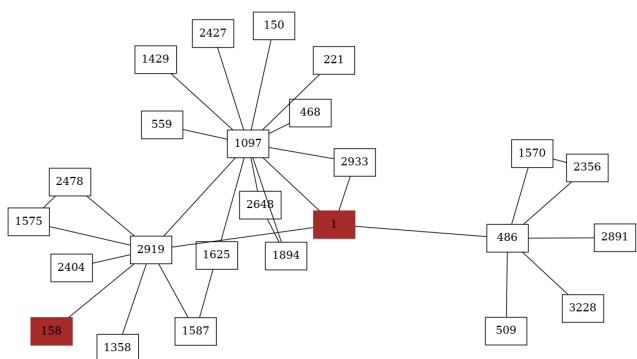

Figure 7: Link-centered subgraph visualization with "Unique" labeling scheme.

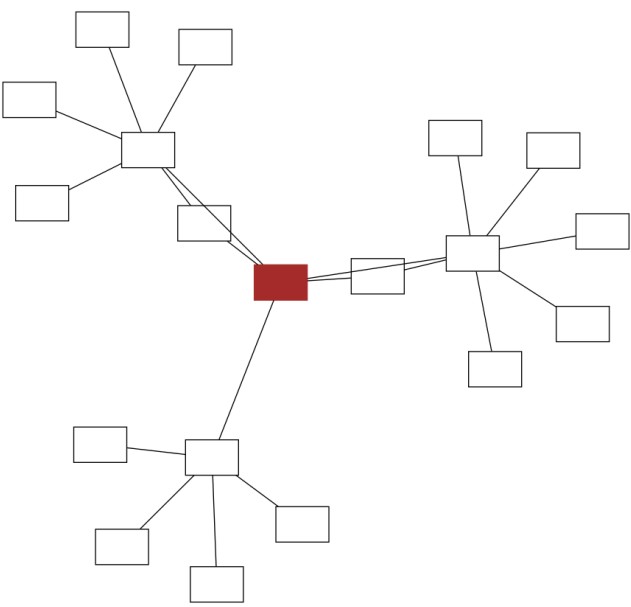

Figure 8: Node-centered subgraph visualization with "No-label" labeling scheme.

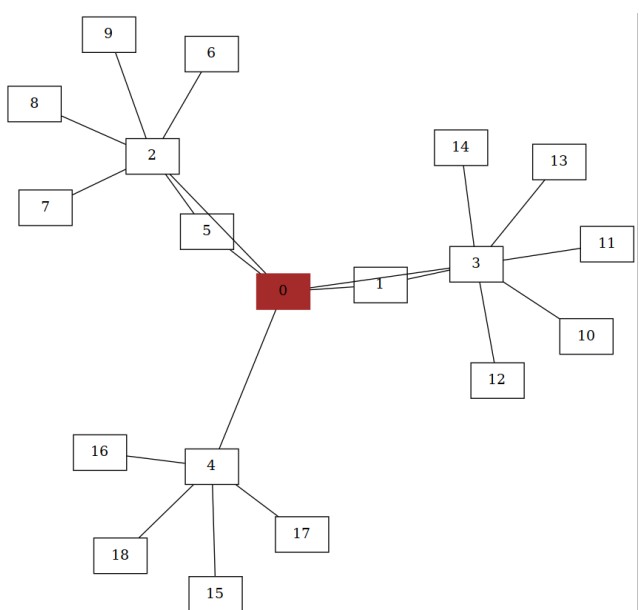

Figure 9: Node-centered subgraph visualization with "Re-label" labeling scheme.

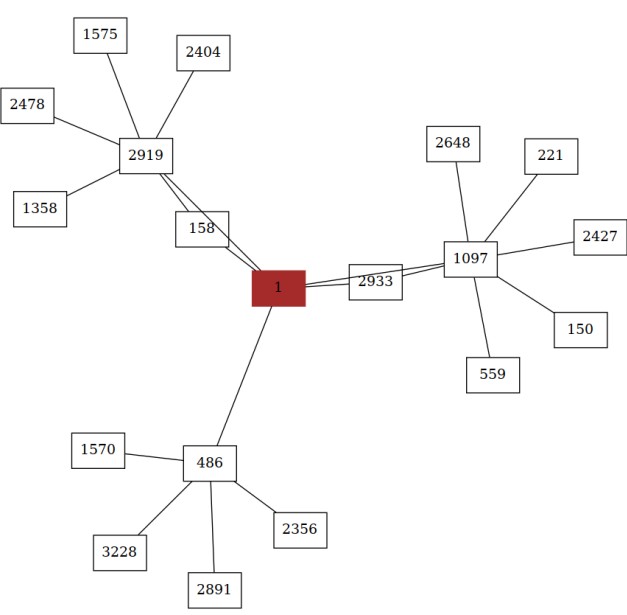

Figure 10: Node-centered subgraph visualization with "Unique" labeling scheme.

Figures 6 and 9 adopt the "Re-label" labeling scheme, where the nodes within the subgraph are reassigned labels starting from 0. This local relabeling introduces some OCR noise, compelling the model to be more robust.

Finally, in Figures 7 and 10, which apply the "Unique" labeling scheme, nodes are labeled with their original IDs from the dataset. This might leverage the OCR capability to match nodes across various subgraphs due to the unique identifiers, which is beneficial for identifying node correspondences. However, this method may hamper generalizability and expose the model to the long-tail problem, where the model's performance degrades for nodes that appear infrequently in the data.

## G   THE PRE-TRAINED MODEL WEIGHT WE UTILIZED

Here we list the link to the pre-trained model weight utilized in this paper:

- ResNet50: https://download.pytorch.org/models/resnet50-0676ba61.pth.

- VGG16: https://download.pytorch.org/models/vgg16-397923af.pth.

- ViT: huggingface.co/facebook/deit-base-patch16-224/resolve/main/pytorch_model.bin.

## H   THE EFFECT OF COLORS IN IMAGE REPRESENTATION

In this section, we explore the effects of different color choices for node representations in graph images.

We first altered the colors of central nodes while keeping surrounding nodes white and evaluated performance on the Cora and Citeseer datasets (Hits@100). The results are summarized in Table 9.

Table 9: Performance (Hits@100) with Different Central Node Colors

| Center Node | GVN-L (Cora) | GVN-N (Cora) | GVN-L (Citeseer) | GVN-N (Citeseer) |
|---|---|---|---|---|
| Black | **90.72±0.52** | 91.43±0.31 | **94.12±0.58** | **94.46±0.52** |
| Brown | 90.70±0.56 | **91.47±0.36** | **94.12±0.58** | 94.44±0.53 |
| Dark Blue | 90.71±0.48 | 91.45±0.44 | 94.09±0.45 | 94.39±0.47 |
| Red | 90.66±0.50 | 91.40±0.40 | 94.00±0.50 | 94.30±0.50 |
| Green | 90.60±0.55 | 91.35±0.45 | 93.95±0.55 | 94.25±0.55 |
| Yellow | 90.68±0.57 | 91.42±0.38 | 94.05±0.57 | 94.40±0.54 |
| White | 89.35±0.72 | 89.90±0.65 | 93.20±0.72 | 93.55±0.75 |

From the above results, we have several findings:

**Findings 1** Only slight differences in performance when the model could distinguish central nodes from surrounding nodes. However, the model showed a preference for darker colors.

**Findings 2** When central nodes became white (indistinguishable from others), there was a noticeable performance degradation. This highlights the significance of labeling the identification of center nodes.

To further illustrate Findings 2, we assigned colors to the nodes surrounding the central nodes. The results are presented in Table 10.

Table 10: Performance (Hits@100) with Different Surrounding Node Colors

| Center Node | Surrounding Node | GVN-L (Cora) | GVN-N (Cora) | GVN-L (Citeseer) | GVN-N (Citeseer) |
|---|---|---|---|---|---|
| Black | Black (same color) | 89.00±0.60 | 89.50±0.55 | 93.00±0.65 | 93.40±0.60 |
| White | White (same color) | 89.35±0.72 | 89.90±0.65 | 93.20±0.72 | 93.55±0.75 |
| Black | Brown (near color) | 90.20±0.50 | 90.70±0.45 | 93.80±0.55 | 94.10±0.50 |
| Black | White (opposite color) | **90.72±0.52** | **91.43±0.31** | **94.12±0.58** | **94.46±0.52** |

These results further reflect the preference of the model for more pronounced color differences between central and surrounding nodes, as indicated in Findings 2. The performance is lower when colors are the same or similar, and higher when there is a clear distinction.

## I    THE EFFECT OF NODE SHAPES IN IMAGE REPRESENTATION

In this section, we investigate the impact of different node shapes on model performance. We experimented with three different shapes: Box, Circle, and Ellipse.

Table 11: Performance (Hits@100) with Different Node Shapes

| Center Node | GVN-L (Cora) | GVN-N (Cora) | GVN-L (Citeseer) | GVN-N (Citeseer) |
|---|---|---|---|---|
| Box | 90.70±0.56 | **91.47±0.36** | 94.12±0.58 | 94.44±0.53 |
| Circle | 90.65±0.52 | 91.40±0.38 | **94.15±0.43** | 94.42±0.50 |
| Ellipse | **90.72±0.46** | 91.45±0.44 | 94.10±0.57 | **94.46±0.52** |

According to Table 11, we find there is no obvious preference for a particular node shape, which finding is aligned with similar observations in GITA (Wei et al., 2024).

## J    TRAINING VISION ENCODER FOR GRAPH STRUCTURE RECONSTRUCTION

This section explores the potential benefits of training the vision encoder to reconstruct graph structures. We introduce an additional training phase for the ResNet50 vision encoder, where it learns to predict the existence of masked edges (i.e., link prediction) in corresponding subgraphs.

Table 12: Performance (Hits@100) with and without Reconstructed Vision Encoder

| Model | Cora | Citeseer |
|---|---|---|
| GVN-L | 90.70±0.56 | 94.12±0.58 |
| GVN-L + Reconstructed VE | 90.68±0.61 | 94.19±0.31 |
| GVN-N | 91.47±0.36 | 94.44±0.53 |
| GVN-N + Reconstructed VE | 91.50±0.47 | 94.46±0.54 |

With Table 12, we find that the performance improvements from this practice are marginal. This may be because the pre-trained ResNet50 is already robust enough to capture various abstract levels of textures within the graph and reflect them in its feature dimensions. Moreover, the visualized subgraphs are structurally clear, without complex backgrounds and distractions, reducing the difficulty of tasks and making such an extra separate training stage becomes unnecessary. Therefore, considering the computational overhead and complexity introduced by this additional training stage, we still recommend using the original version of GVN.

## K    ABLATION STUDY: MAPPING MATRIX IN CROSS-ATTENTION

Table 13: Performance (Hits@100) with and without Mapping Matrices in Cross-Attention

| Model | Cora (Hits@100) | Citeseer (Hits@100) |
|---|---|---|
| GVN-L | 90.70±0.56 | 94.12±0.58 |
| GVN-L + Mapping Matrices | 90.81±0.60 | 94.05±0.62 |
| GVN-N | 91.47±0.36 | 94.44±0.53 |
| GVN-N + Mapping Matrices | 91.35±0.40 | 94.50±0.55 |

This section presents an ablation study to evaluate the impact of removing the mapping matrices $W_Q$, $W_K$, and $W_V$ in the cross-attention mechanism, which are typically used to project input features into query, key, and value spaces.

We conducted experiments on the Cora and Citeseer datasets to assess the performance implications of excluding these mapping matrices.

The results in Table 13 indicate that removing the mapping matrices $W_Q$, $W_K$, and $W_V$ does not impact the model's effectiveness. Therefore, we omit them in GVN to make the framework more concise.

