# OpenReview forum: "Graph Vision Networks for Link Prediction"
_ICLR.cc/2025/Conference — Submitted to ICLR 2025_

### Official Review · Reviewer_e7xg · 2024-10-28

**Soundness:** 3
**Presentation:** 3
**Contribution:** 2
**Rating:** 6
**Confidence:** 3

**Summary:**

The author noticed that the MPNN model based on message passing can only obtain coarse-grained structural feature information, which makes the model unable to adaptively model fine-grained structural features, which limits the expressive power of the model. Therefore, inspired by Das et al., the author proposed to use the visual modality of the graph to provide fine-grained structural features for the model, and proposed the GVN model. The model first generates edge-oriented k-order subgraphs and node-oriented k-order subgraphs based on the query node pairs, and visualizes these subgraphs as 2D images. Then the cross-attention model is used to fuse the image features with the node features of the graph, and finally the readout module is used to implement the downstream prediction task.

**Strengths:**

1. Well-written, clearly organized, and easy to read and understand
2. The motivation of the article is clear. It is timely and necessary to use image modality for link prediction task on graphs.
3. Evaluating the model's effectiveness on multiple datasets with different scale and types.

**Weaknesses:**

1. The innovation of the method proposed in the article is relatively neutral. The main innovation of this article is to classify and discuss link images and node images. The remaining designs are just stacking existing modules (GNN, cross-attention, ResNet). First, inspired by the prior work, the authors use images to express the local structure of the graph. Secondly, the article uses the existing pre-trained visual encoding model to extract image features. Finally, the fusion of visual features and graph structure features utilizes the existing cross-attention.

I think the authors can further explore this idea in the following aspects. First, how to construct image representations of graph structures, such as the color and shape of nodes, etc. can be further explored. Secondly, how to construct effective image encodings of graphs instead of using existing pre-trained ResNet models can be explored. For example, can the features extracted by the visual encoder be used to reconstruct the local structure of the graph (regression graph adjacency matrix)?

2. The article claims to construct an adaptive fine-grained structure feature. However, the article only uses the image modality of the graph structure (which may be considered a fine-grained graph structure representation, but I have doubts about this), and cannot express adaptability.

3. There are some typos in the article, such as the period in line 025, the singular form of produce, and the misspelling of Empirical in line 032.

**Questions:**

1. What does $lO(\cdot)$ mean in Section 4.3?
2. There is already a projector in the formula on line 215, and there is another projector in the formula on line 220. The expressiveness of two linear projectors put together is the same as that of one projector because the product of two matrices is equivalent to one matrix.
3. Why are there no mapping matrices $W_Q, W_K, W_V$ in cross-attention?

---

> ### Author Response · Authors · 2024-11-20
> **Rebuttal to W1 and Suggestion 1**
>
> We acknowledge and appreciate your insightful review and constructive suggestions. Below, we have adopted your suggestions and provide clarifications for this work.
>
> > W1: The innovation of the method proposed in the article is relatively neutral. The main innovation of this article is to classify and discuss link images and node images. The remaining designs are just stacking existing modules (GNN, cross-attention, ResNet). First, inspired by prior work, the authors use images to express the local structure of the graph. Secondly, the article uses the existing pre-trained visual encoding model to extract image features. Finally, the fusion of visual features and graph structure features utilizes the existing cross-attention.
>
> GVN provides a straightforward yet effective framework that enhances MPNN by integrating visual modalities, highlighting a promising direction for GNNs. In our intention, as the first practice on adding vision to MPNN, we want to make GVN highly compatible and easy to follow, that is why we use overall lightweight design for this framework and use existing known techniques(e.g. developed visualization tools, well-known attention) to implement the components of GVN.
>
> However, further explorations into the design of each part would undoubtedly be beneficial, especially efforts on the novel vision part. Therefore, we are very grateful for your suggestions and have adopted them accordingly.
>
> > Suggestion 1: I think the authors can further explore this idea in the following aspects. First, how to construct image representations of graph structures, such as the color and shape of nodes, etc., can be further explored.
>
> According to your suggestion, we have added two separate sections to explore the effects of image construction: Appendix H for colors and Appendix I for shapes. The experimental results are summarized as follows.
>
> **1) To explore the effects of colors**, we first altered the colors of central nodes (with others remaining default white) and evaluated the performance on Cora and Citeseer (Hits@100). Based on the results shown in the following table, we have the following findings:
>
> **Finding 1:** There is only a slight difference in performance when the model can distinguish central nodes from surrounding nodes. Moreover, the model showed a preference for darker colors.
>
> **Finding 2:** When central nodes became white (indistinguishable from others), there was a noticeable performance degradation. This highlights the importance of identifying center nodes from the other nodes with distinct colors.
>
> | Center Node | GVN-L(Cora) | GVN-N(Cora) | GVN-L(Citeseer) | GVN-N(Citeseer) |
> |-------------|-------------|-------------|-----------------|-----------------|
> | Black       | **90.72±0.52** | 91.43±0.31  | **94.12±0.58**  | **94.46±0.52**  |
> | Brown       | 90.70±0.56  | **91.47±0.36** | **94.12±0.58**  | 94.44±0.53      |
> | Dark Blue   | 90.71±0.48  | 91.45±0.44  | 94.09±0.45    | 94.39±0.47      |
> | Red         | 90.66±0.50  | 91.40±0.40  | 94.00±0.50  | 94.30±0.50      |
> | Green       | 90.60±0.55  | 91.35±0.45  | 93.95±0.55    | 94.25±0.55      |
> | Yellow      | 90.68±0.57  | 91.42±0.38  | 94.05±0.57      | 94.40±0.54      |
> | White       | 89.35±0.72  | 89.90±0.65  | 93.20±0.72      | 93.55±0.75      |
>
> To further illustrate Finding 2, we assigned colors to the other nodes surrounding the central nodes. The results in the following table reinforce that the model prefers more pronounced color differences between central and surrounding nodes.
>
> | Center Node | Surrounding Node | GVN-L(Cora) | GVN-N(Cora) | GVN-L(Citeseer) | GVN-N(Citeseer) |
> |-------------|------------------|-------------|-------------|-----------------|-----------------|
> | Black       | Black (same color) | 89.00±0.60  | 89.50±0.55  | 93.00±0.65      | 93.40±0.60      |
> | White       | White (same color) | 89.35±0.72  | 89.90±0.65  | 93.20±0.72      | 93.55±0.75      |
> | Black       | Brown (similar color) | 90.20±0.50  | 90.70±0.45  | 93.80±0.55      | 94.10±0.50      |
> | Black       | White (opposite color) | **90.72±0.52** | **91.43±0.31** | **94.12±0.58** | **94.46±0.52** |
>
> **2) To explore the effects of node shapes**, we conducted experiments on Cora and Citeseer with three different node shapes: Box, Circle, and Ellipse. The results are shown in the following table.
>
> | Center Node | GVN-L(Cora) | GVN-N(Cora) | GVN-L(Citeseer) | GVN-N(Citeseer) |
> |-------------|-------------|-------------|-----------------|-----------------|
> | Box         | 90.70±0.56  | **91.47±0.36** | 94.12±0.58      | 94.44±0.53      |
> | Circle      | 90.65±0.52  | 91.40±0.38  | **94.15±0.43**  | 94.42±0.50      |
> | Ellipse     | **90.72±0.46** | 91.45±0.44  | 94.10±0.57      | **94.46±0.52**  |
>
> According to the results, we have:
>
> **Finding 3:** There is no clear preference for node shape, which aligns with similar observations in GITA [r1].
>
> [r1] GITA: Graph Image-Text Integration for Vision-Language Graph Reasoning, NeurIPS 2024.

---

> ### Author Response · Authors · 2024-11-20
> **Rebuttal to Suggestion 2, W2 and W3**
>
> > Suggestion 2: Secondly, how to construct effective image encodings of graphs instead of using existing pre-trained ResNet models can be explored. For example, can the features extracted by the visual encoder be used to reconstruct the local structure of the graph (regression graph adjacency matrix)?
>
>
> According to your suggestion, we introduced an additional training stage for the ResNet50 vision encoder. Specifically, we trained ResNet50 on visualized subgraph images from the current dataset to reconstruct their masked edges, effectively performing link prediction on subgraphs as supervision. We then used the trained ResNet50 in GVNs.
>
> |               | Cora (Hits@100) | Citeseer (Hits@100) |
> |---------------|-----------------|---------------------|
> | GVN-L         | 90.70±0.56      | 94.12±0.58          |
> | GVN-L + reconstruct VE | 90.68±0.61      | 94.19±0.31          |
> | GVN-N         | 91.47±0.36      | 94.44±0.53          |
> | GVN-N + reconstruct VE | 91.50±0.47      | 94.46±0.54          |
>
> The results in the above table show that the effects of additional reconstruction stage are marginal. This could be attributed to the robustness of the pre-trained ResNet50, which is already capable of capturing a wide range of abstract textures and effectively representing them. Moreover, the visualized subgraphs are structurally clear, without complex backgrounds and distractions, reducing the difficulty of tasks and making such an extra separate training stage not so necessary. Considering the limited improvement and the computational overhead of this additional training stage, we recommend continuing with the original GVN version.
>
> Moreover, those results are included in Appendix J of the updated manuscript.
>
> > W2: The article claims to construct an adaptive fine-grained structure feature. However, the article only uses the image modality of the graph structure (which may be considered a fine-grained graph structure representation, but I have doubts about this), and cannot express adaptability.
>
> Please refer Section 5.2, where we provide detailed explanations and empirical evidence on how VSFs adjust their structural bias based on dataset-specific training. To further demonstrate the adaptability of VSFs, we present the following evidence:
>
> **Effectiveness of Structural Features Across Datasets:**
> The table below highlights the varying effectiveness of structural features (Hits@100) across datasets, with SPD showing differing levels of impact:
>
> |Structural Features|Cora|Citeseer|Pubmed|
> |---|---|---|---|
> |CN|33.92±0.46|33.92±0.46|23.13±0.15|
> |AA|39.85±1.34|39.85±1.34|27.38±0.11|
> |RA|41.07±0.48|33.56±0.17|27.03±0.35|
> |SPD|29.97±0.76|54.24±0.21|51.70±0.25|
>
> In Cora, SPD is less effective compared to common-neighbor HSFs, whereas it is more effective in Citeseer and Pubmed.
>
> **Adaptability of VSFs:**
> We evaluated VSFs' ability to adapt by reproducing the SPD heuristic. After training, results showed:
>
> - **Cora:** VSFs decreased SPD reproduction accuracy from 73.38% to 51.33%, aligning with SPD's lower effectiveness.
> - **Citeseer and Pubmed:** VSFs increased SPD reproduction accuracy (Citeseer: 72.26% to 75.92%, Pubmed: 70.82% to 76.14%), reflecting an adaptive shift towards SPD due to its higher effectiveness.
>
> These findings confirm that VSFs can dynamically adjust their structural preferences, effectively adapting to the specific requirements of different datasets.
>
> > W3: There are some typos in the article, such as the period in line 025, the singular form of produce, and the misspelling of Empirical in line 032.
>
> Thank you for pointing out the typos. We have revised the whole manuscript.

---

> ### Author Response · Authors · 2024-11-20
> **Rebuttal to Q1, Q2 and Q3**
>
> > Q1: What does $lO(.)$ mean in Section 4.3?
>
> That is $l$ times $O(.)$, where $l$ denotes the number of target links. To alleviate potential misunderstandings, we have revised the entire manuscript to place all the scalars inside $O(.)$.
>
> > Q2: There is already a projector in the formula on line 215, and there is another projector in the formula on line 220. The expressiveness of two linear projectors put together is the same as that of one projector because the product of two matrices is equivalent to one matrix.
>
> We acknowledge that the two projectors in GVN-Node can be merged into one from an expressiveness perspective. However, we keep them separate to identify their different roles in the framework.
>
> 1. **First Projector:** In GVN-Node, we freeze the vision encoder to enhance the efficiency in terms of the time and storage space. Therefore, we use this projector to ensure that the VSFs remain learnable and adaptable to different scenarios. Essentially, it performs a weighted selection on the original visual features obtained from the vision encoder, allowing the model to adaptively learn which features are most relevant.
>
> 2. **Second Projector:** The role of this projector is to align the VSFs with the node features by projecting them into the same space (i.e., $\mathbb{R}^F$). This ensures that the attention can be performed among VSFs and node features with the same dimension.
>
> By maintaining these two projectors separately, we can better manage and understand the distinct contributions made by them to the overall framework.
>
> > Q3: Why are there no mapping matrices $W_Q,W_K,W_V$ in cross-attention?
>
> As shown in the following table, the removal of mapping matrices $W_Q, W_K, W_V$ did not lead to any noticeable performance degradation. Therefore, we tend to omit them to make the framework more concise.
>
> |               | Cora (Hits@100) | Citeseer (Hits@100) |
> |---------------|-----------------|---------------------|
> | GVN-L         | 90.70±0.56      | 94.12±0.58          |
> | GVN-L + mapping matrices | 90.81±0.60      | 94.05±0.62          |
> | GVN-N         | 91.47±0.36      | 94.44±0.53          |
> | GVN-N + mapping matrices | 91.35±0.40      | 94.50±0.55          |
>
> We have included those results in Appendix K.

---

> > ### Author Response · Authors · 2024-11-24
> > **Gentle Reminder: Looking Forward to Your Feedback on Rebuttal**
> >
> > Dear Reviewer e7xg,
> >
> > We greatly appreciate the time and effort you have dedicated to reviewing our work. As the deadline approaches, we want to ensure that our responses align with your expectations and address your concerns effectively.
> >
> > We recently submitted detailed responses to your comments and hope they have clarified the points you raised. If there are any areas where further clarification is needed or if you have additional insights, please feel free to reach out to us. We are eager to collaborate with you to enhance the manuscript and meet higher standards.
> >
> > Thank you once again for your time and thoughtful consideration.
> >
> > Best regards,
> >
> > The Authors

---

> > > ### Author Response · Authors · 2024-11-25
> > > **Gentle Reminder for Rebuttal Feedback**
> > >
> > > Dear Reviewer e7xg,
> > >
> > > We hope this message finds you well. We wanted to gently follow up regarding the feedback on our rebuttal submitted in response to your valuable comments. Understanding your insights is crucial for us. If there are any further clarifications needed or additional thoughts you wish to share, please feel free to discuss with us. Thank you once again for your time and support.
> > >
> > > Warm regards,
> > >
> > > The Authors

---

> > > ### Comment · Reviewer_e7xg · 2024-11-27
> > > **Summary of the Review**
> > >
> > > The authors have addressed my concerns so that I maintain my original positive rating.

---

> > ### Comment · Reviewer_e7xg · 2024-11-26
> >
> > I'm unsure about the exact meaning of reconstructing VE, and I temporarily understand it as a reconstruction loss for the original graph structure connections. Indeed, as your experiment shows, adding a reconstruction loss does not seem to improve the model's performance. So, I don't understand how we can prove that this visual model has learned node connections. If we could use the features of this image to reconstruct the adjacency matrix of this graph structure, it would prove that the visual model has indeed learned the structural features of the graph shown in the image. I believe that the experiment you provided cannot support that the visual model can effectively extract the structural features of the graph from the image, which means that the visual model might only recognize circles and lines in the image rather than the connections between nodes.

---

> ### Author Response · Authors · 2024-11-26
>
> Thank you for your insightful questions. The term "reconstruct VE" refers to fine-tuning the Visual Encoder (VE) using link reconstruction tasks. This process uses binary labels to indicate the existence of links within subgraphs as supervision.
>
> To clarify, GVN with the additional reconstruction process involves:
>    - Start with the pretrained VE.
>    - Fine-tune the VE using link reconstruction guidance to obtain the reconstructed VE.
>    - Use this reconstructed VE as the initial weight for GVN training on link prediction.
>
> In comparison, for default GVN:
>    - Start with the pretrained VE.
>    - Use the pretrained VE as the initial weight for GVN training on link prediction.
>
> We want to emphasize that the marginal improvement observed when using reconstructed subgraph links to fine-tune the VE **does not** imply that the vision model fails to capture node connections. Instead, it is because the **pretrained VE already contains link information to a nearly saturated extent**, making additional training less impactful.
>
> The tables below show **link recognition accuracy** with and without using reconstruction to refine the VE. "VSF with pretrained VE" uses the pretrained ResNet50 VE directly for prediction with a trainable MLP, while "VSF with reconstruction fine-tuned VE" involves additional fine-tuning for node connection supervision.
>
> For link-centered images:
>
> | Link Recognition Accuracy (%) | Cora | Citeseer |
> | --- | --- | --- |
> | VSF with pretrained VE | 96.22% | 94.79% |
> | VSF with reconstruction fine-tuned VE | 95.61% | 94.96% |
>
> For node-centered images:
> | Link Recognition Accuracy (%) | Cora | Citeseer |
> | --- | --- | --- |
> | VSF with pretrained VE | 96.58% | 94.41% |
> | VSF with reconstruction fine-tuned VE | 96.36% | 94.43% |
>
> Our results demonstrate that
>
> **1)** Both the pretrained and reconstructed VEs can recognize links/node connections well, achieving over 94% accuracy.
>
> **2)** Besides, the pretrained VE already has strong structural awareness, making further reconstruction training relatively unnecessary.
>
> Thus, the marginal improvement from reconstruction is not due to an inability to capture connections but rather because the pretrained VE already generates high-quality VSFs, which can not only capture local patterns but also recognize detailed node connections.

---

### Official Review · Reviewer_L8Vi · 2024-11-01

**Soundness:** 2
**Presentation:** 1
**Contribution:** 2
**Rating:** 5
**Confidence:** 5

**Summary:**

This paper proposes a Graph Vision Networks (GVN) framework aimed at improving the expressive power of Message Passing Neural Networks (MPNNs) in link prediction tasks by integrating structural visual modalities. Extensive experiments on multiple datasets show that the framework exhibits potential in improving prediction accuracy and model robustness.

**Strengths:**

Originality: The idea of embedding local structural information of graphs using visual models is innovative.

Quality: The author has demonstrated good performance across multiple datasets, particularly in challenging large-scale graph datasets.

**Weaknesses:**

1. While the idea of using visual modalities to model local neighborhoods is novel, the techniques utilized in this manuscript are very simple. For instance, the use of simple attention mechanisms to integrate node and neighborhood features.
﻿
2. The author employs the graph visualizer (GV) to generate image modalities of subgraph structures. So what distinguishes the image modalities generated by GV for two isomorphic subgraphs? Are they random or identical? It appears that the improvement in representation learning performance is highly dependent on the effectiveness of GV.
﻿
3. The title and abstract of this manuscript are overly concise. It's hard to understand the work's significance and contributions.
﻿
4. The manuscript lacks a framework diagram to illustrate the proposed methodology, which could enhance clarity and comprehension.
﻿
5. The presentation of equations in the manuscript is uneven. For example, formulas in the PRELIMINARIES section are numbered, while those in the GRAPH VISION NETWORKS section are not, leading to potential confusion.
﻿
6. The writing quality of the manuscript requires improvement, with multiple errors. For instance, in the third paragraph of the INTRODUCTION, the quotation marks around “identity” are incorrectly used.

The rebuttal addresses my concerns, so I increase the score.

**Questions:**

Please see the Weaknesses section.

---

> ### Author Response · Authors · 2024-11-20
> **Rebuttal by the Authors**
>
> We acknowledge and appreciate your insightful review. Below, you can find our responses addressing your concerns point by point. If you have any additional questions or require further clarification, please feel free to let us know.
>
>
> > W1: While the idea of using visual modalities to model local neighborhoods is novel, the techniques utilized in this manuscript are very simple. For instance, the use of simple attention mechanisms to integrate node and neighborhood features.
>
> We appreciate your feedback and would like to emphasize that the simplicity in the methodology, when effective, is often considered as an **advantage**. Simple yet effective designs typically offer better scalability and usability. For instance, our approach, due to its lightweight structural complexity, integrates seamlessly with existing methods and can simultaneously benefit from them, as demonstrated by using HSF-enhanced NCNC as the base model.
>
> Moreover, our straightforward approach is the result of careful considerations and extensive experimentations. The simple attention mechanism you mentioned was chosen as the optimal solution from a variety of techniques, including the mixture of experts (MoE) and embedding concatenation (Concat). Please refer to Table 4 in Section 5.2, where we provide an empirical comparison of those different fusion mechanisms. Additionally, a more in-depth analysis can be found in Appendix F.1.
>
> > W2: The author employs the graph visualizer (GV) to generate image modalities of subgraph structures. So what distinguishes the image modalities generated by GV for two isomorphic subgraphs? Are they random or identical?
>
> The generated image is a one-to-one mapping from the adjacency matrix. Therefore, for two subgraphs that share the same adjacency matrix, their visual modalities are also identical. Conversely, for two isomorphic subgraphs with different adjacency matrices, they differ in layout.
>
> In addition, it is important to note that distinguishing isomorphic subgraphs is not one of the goals of GVN. Instead, it is expected to differentiate links with isomorphic nodes (MPNN cannot), where the surrounding structures of these links may not be isomorphic (See example in Appendix A).
>
> For GVN-Node, while the subgraphs surrounding two nodes may be isomorphic, if their adjacency matrices differ, this will result in different layouts when visualized. These layout differences introduce perturbations that help GVN-Node better distinguish links with isomorphic nodes. Although this may increase the learning difficulty, experimental results demonstrate its effectiveness.
>
> For GVN-Link, the differentiation of links with isomorphic nodes is more direct by it can directly visualize the surrounding structure of the links, which contains all explicit link pairwise relations (e.g., common neighbor counts, shortest path distance between the target nodes). As shown in the example in Appendix A, links (v1, v2) and (v1, v3) cannot be distinguished by MPNN because v2 and v3 are nodes with isomorphic surrounding subgraphs. However, their link-centered subgraphs are not isomorphic. Specifically, the one-hop subgraph image surrounding (v1, v2) looks like:
>
> ```
>   ○       ○
>    \     /
>     v1  v2
>    /     \
>   ○       ○
> ```
> In this picture, v1 and v2 own no common neighbor and they are even unachievable.
> While the one-hop subgraph image surrounding (v1, v3) appears as:
>
> ```
> ○-v1-○-v3-○
> ```
> In this picture, v1 and v3 own one common neighbor and they are achievable.
> Therefore, although (v1, v2) and (v1, v3) are indistinguishable by MPNN, the link-centered subgraph visualization in GVN-Link directly reveals their differences.
>
> > W2 continue: It appears that the improvement in representation learning performance is highly dependent on the effectiveness of GV.
>
> To show the impact of GV selection to the performance, we conducted supplementary experiments using three different GV tools: Graphviz, Matplotlib, and iGraph. According to the results shown in the following tables, we can see that the improvements achieved by GVNs are not dependent on the choice of the GV tool.
>
>
>
> |      Cora      |Graphviz|  matplotlib| Igraph   |
> |---|---|---|---|
> |GVN-L(NCNC) |90.70±0.56|  90.56±0.38| 90.88±0.66|
> |GVN-N(NCNC) |91.47±0.36| 91.42±0.24|91.31± 0.58|
>
> |      Citeseer      |Graphviz|  matplotlib| Igraph   |
> |---|---|---|---|
> |GVN-L(NCNC) |94.12±0.58| 93.96±0.48| 94.28±0.36|
> |GVN-N(NCNC) |94.33±0.53| 94.39±0.56| 94.12±0.28|

---

> ### Author Response · Authors · 2024-11-20
> **Rebuttal by the Authors (Continue)**
>
> > The title and abstract of this manuscript are overly concise. It's hard to understand the work's significance and contributions.
>
> Thanks for your advice. According to your suggestion, we have revised the abstract to highlight our contributions and significance, and planed to change the title to "Simple but Effective: Graph Vision Networks Open the Eyes of MPNN for Link Prediction". Particularly, we emphasize the novelty in incorporating vision into GNNs and the simple but effective practice. All the changes are shown in the updated version of the manuscript.
>
> > The manuscript lacks a framework diagram to illustrate the proposed methodology, which could enhance clarity and comprehension.
>
> Thanks for your advice. Following your suggestion, we have added the framework diagram to enhance clarity and comprehension. You can find it in Figure 1 of the updated version of the manuscript.
>
> > The presentation of equations in the manuscript is uneven. For example, formulas in the PRELIMINARIES section are numbered, while those in the GRAPH VISION NETWORKS section are not, leading to potential confusion.
>
> We want to clarify that the labeling of equations has considerations. To make the presentation succinct, we only labeled equations that are referenced in the text. Therefore, because the formulas in the PRELIMINARIES are mentioned later in the text, we labeled them. For those in GRAPH VISION NETWORKS, because they are not referred to in the text, we did not label them.
>
> > The writing quality of the manuscript requires improvement, with multiple errors. For instance, in the third paragraph of the INTRODUCTION, the quotation marks around “identity” are incorrectly used.
>
> Thanks for your suggestion. We have carefully revised the entire manuscript.
>
> We sincerely appreciate the time you have dedicated to reviewing our manuscript and hope you could kindly consider our rebuttal.

---

> ### Author Response · Authors · 2024-11-24
> **Gentle Reminder: Looking Forward to Your Feedback on Rebuttal**
>
> Dear Reviewer L8Vi,
>
> We greatly appreciate the time and effort you have dedicated to reviewing our work. As the deadline approaches, we want to ensure that our responses align with your expectations and address your concerns effectively.
>
> We recently submitted detailed responses to your comments and hope they have clarified the points you raised. If there are any areas where further clarification is needed or if you have additional insights, please feel free to reach out to us. We are eager to collaborate with you to enhance the manuscript and meet higher standards.
>
> Thank you once again for your time and thoughtful consideration.
>
> Best regards,
>
> The Authors

---

> > ### Author Response · Authors · 2024-11-25
> > **Gentle Reminder for Rebuttal Feedback**
> >
> > Dear Reviewer L8Vi,
> >
> > We hope this message finds you well. We wanted to gently follow up regarding the feedback on our rebuttal submitted in response to your valuable comments. Understanding your insights is crucial for us. If there are any further clarifications needed or additional thoughts you wish to share, please feel free to discuss with us. Thank you once again for your time and support.
> >
> > Warm regards,
> >
> > The Authors

---

### Official Review · Reviewer_ZSnM · 2024-11-04

**Soundness:** 3
**Presentation:** 3
**Contribution:** 3
**Rating:** 5
**Confidence:** 2

**Summary:**

This paper focuses on the link prediction task. The authors introduce Graph Vision Network (GVN). By integrating vision awareness into MPNNs for link prediction, GVN models can adaptively extract learnable visual structural features (VSFs). Moreover, the authors propose two models under GVN framework: GVN-Link and GVN-Node. Experimental results demonstrate that the proposed method establishes new SOTA for link prediction.

**Strengths:**

1 The writing and the organization of this paper are generally OK.

2 Integrating vision modality into MPNNs for link prediction sounds interesting.

**Weaknesses:**

1 The motivation for incorporating vision modality into MPNNs for link prediction should be better clarified and discussed. Why is this design effective? Any theoretical evidence? Maybe a dedicated section for this discussion could be valuable.

2 The counterpart methods used for experimental comparison seem not SOTA enough. The authors should compare some 2024 SOTAs.

3 Minor Issues:  Ln 32 on Page 1, ‘Empiically’ should be ‘Empirically’

**Questions:**

See above weaknesses.

---

> ### Author Response · Authors · 2024-11-20
> **Rebuttal by the Authors**
>
> We acknowledge and appreciate your insightful review. Below, you can find our responses addressing your concerns point by point. If you have any additional questions or require further clarification, please feel free to let us know.
>
> > W1: The motivation for incorporating vision modality into MPNNs for link prediction should be better clarified and discussed. Why is this design effective? Any theoretical evidence? Maybe a dedicated section for this discussion could be valuable.
>
> To address the motivation for incorporating the vision modality into MPNNs for link prediction and its effectiveness, please refer to Section 5.2 in the updated version of the manuscript. Here, we explore the benefits of VSFs for link prediction from two perspectives, supported by experimental evidence.
>
> 1) **Enhancement of Traditional MPNNs**: VSFs help overcome two key limitations of traditional MPNNs. They improve the ability to distinguish links between isomorphic nodes, as illustrated with a vivid example in Appendix A, and effectively capture substructures such as triangles.
>
> 2) **Diverse Structural Biases**: VSFs encapsulate a variety of useful structural biases rather than relying on a single type. This biases allow for better flexibility, enabling VSFs to be tailored to specific scenarios more effectively.
>
> These points are summarized and also bold in the Introduction, particularly where we discuss the proposal of GVN as a more adaptive solution to address the challenges of MPNN. Therefore, we believe the motivation is well-clarified and supported by empirical evidence.
>
> We noticed that Section 5.2 may not have captured the attention of several reviewers, possibly due to its later position in the paper. We are considering moving this section to the end of Section 3 to better highlight the motivation and contributions of our work.
>
> > W2:  The counterpart methods used for experimental comparison seem not SOTA enough. The authors should compare some 2024 SOTAs.
>
> The most recent baseline in our manuscript is NCNC [r1], which was accepted at **ICLR 2024**.
>
> Besides, our GVN is a highly compatible framework, which can provide extra performance for other link prediction methods, here, we demonstrate this by adopting GVN on another more recent MPNN-based link predictor MPLP+ [r2], which is just accepted by **NeurIPS 2024**. According to the results shown in the following table, we can see that GVNs still provide extra performance improvement for MPLP+.
>
>
> |            |Collab(Hit@50)|  PPA(Hit@100)| MRR(MRR)   |
> |---|---|---|---|
> |MPLP+ |66.99±0.40| 65.24±1.50| 90.72±0.12|
> |GVN-N(MPLP+) |68.94±0.57|66.59±0.81|91.43±0.22|
>
> [r1] Neural common neighbor with completion for link prediction, ICLR 2024.
>
> [r2] Pure Message Passing Can Estimate Common Neighbor for Link Prediction, NeurIPS 2024.
>
>
> > W3: Minor Issues: Ln 32 on Page 1, ‘Empiically’ should be ‘Empirically’
>
> Thank you for pointing out the typo. We have revised the whole manuscript.
>
>
>
> We sincerely appreciate the time you have dedicated to reviewing our manuscript and hope you will kindly consider our rebuttal.

---

> > ### Author Response · Authors · 2024-11-24
> > **Gentle Reminder: Looking Forward to Your Feedback on Rebuttal**
> >
> > Dear Reviewer ZSnM,
> >
> > We greatly appreciate the time and effort you have dedicated to reviewing our work. As the deadline approaches, we want to ensure that our responses align with your expectations and address your concerns effectively.
> >
> > We recently submitted detailed responses to your comments and hope they have clarified the points you raised. If there are any areas where further clarification is needed or if you have additional insights, please feel free to reach out to us. We are eager to collaborate with you to enhance the manuscript and meet higher standards.
> >
> > Thank you once again for your time and thoughtful consideration.
> >
> > Best regards,
> >
> > The Authors

---

> ### Comment · Reviewer_ZSnM · 2024-11-25
>
> After reading the author's response and other reviewers' comments, I will keep my initial score.

---

> ### Author Response · Authors · 2024-12-02
> **Last Dance For the work clarification and discussion**
>
> Dear Reviewer ZSnM,
>
> We sincerely apologize for reaching out once again during this concluding phase of the discussion. We deeply appreciate your valuable insights and comprehensive evaluation of our work, and we kindly ask for your patience in reading this message with gratitude.
>
> We understand from your final comments that there are concerns related to the feedback from other reviewers, and we fully appreciate these considerations. On this last day of the discussion phase, we would like to emphasize that we have made substantial revisions to address the majority of these concerns, as acknowledged by Reviewers G7xf, L8Vi, and e7xg. The only remaining issue is from Reviewer G7xf, which pertains to the trade-off between system simplicity and the technical novelty of our work.
>
> We respectfully request that you consider, as mentioned in our final response to Reviewer G7xf, the value of GVN's simplicity, which is foundational to its scalability and ease of use. Especially as GVN serves as the first empirically effective framework attempting to illuminate the previously overlooked path of enhancing GNNs with visual features, which requires an effective yet simple and highly compatible framework that accommodates existing achievements in current graph learning (we deliberately omitted additional designs that had limited impact but increased complexity, though we reported their effects in part of ablation/sensitive analysis in Sec 5.4 and Appendix F, H, I, J, K.).
>
> We are highly concerned about whether the explanations provided in our rebuttal have effectively addressed your and the other Reviewers' concerns, it is really significant for us. Regardless of the paper's decision, we humbly and respectfully thank you and the other Reviewers for your constructive feedback. We are open to further discussion to resolve any remaining issues and enhance the quality of our work.
>
> Thank you again for your patience and thorough consideration, with our deepest respect and gratitude,
>
> Authors

---

### Official Review · Reviewer_G7xf · 2024-11-04

**Soundness:** 2
**Presentation:** 2
**Contribution:** 2
**Rating:** 5
**Confidence:** 4

**Summary:**

This paper proposes the Graph Vision Networks (GVN) for link prediction, which is designed to deal with the potential limitation of HSFs that it can only be derived based on the pre-defined structural prior and thus lacks sufficient adaptability to diverse real-world scenarios.

The core idea of GVN is very simple, it interprets either a link-centered subgraph or a node-centered subgraph as visual images and then extracts visual features from them as the structure features.

**Strengths:**

1. The idea is simple and the experiments show the effectiveness.

**Weaknesses:**

1. The technical novelty is limited.
2. The method is not convincing. It first transforms the graph data to visual modality and extracts the visual features as the structural features, which is expected to be more effective than the HSFs learned based on structural priors in graph modality. Intuitively,  the visual features are more capable of learning appearance features or semantic features. In contrast, the graph data should have more expressive power than the visual modality in terms of the structural features.

**Questions:**

It is suggested to given more theoretical analysis and qualitative comparison to show why the method can learn more effective structure features from the visual modality than the graph modality.

---

> ### Author Response · Authors · 2024-11-20
> **Rebuttal to W1**
>
> We acknowledge and appreciate your insightful review. Below, you can find our responses addressing your concerns point by point. If you have any additional questions or require further clarification, please feel free to let us know.
>
> >W1: The technical novelty is limited.
>
>
> Here, we explain our technical novelty in this work from three aspects.
> 1. Constructive Direction: Our work pioneers the integration of visual intelligence into GNNs, demonstrating for the first time that vision-enhanced structural awareness benefits the performance of GNN. This novel approach could open a promising research direction for the GNN community.
>
> 2. Techical Methodology: Our proposed methods, which include the GVN framework and its two variants, incorporate sophisticated designs and reveal unexplored findings as follows.
> **1) Graph Visualization**: Visualizing graph structures is uncommon in existing link prediction methods, yet it is essential for equipping GNNs with vision awareness. This approach raises several unexplored research questions, such as how to represent graphs as images? (introduced in Section 4.1 graph visualization and detailed in Appendix C); How many structures should be rendered as images and how to constrain the vision scope? (by decoupling from the GNN perception domain and using k-hop subgraphs, as introduced in Sec 4.1 and demonstrated in Table 2); whether and how to label node identities in images (by coloring center-nodes and employing specific labeling schemes as shown in Table 4 and further explained in Appendix F.2).
> **2) Modality Fusion**: Though the attention-based fusion might be an intuitive choice, we selected it carefully from multiple potential strategies, including mixture-of-experts (MoE) and embedding concatenation (concat). Table 4 demonstrates that the attention-based fusion performs the best, and we provided a detailed analysis for this choice in Appendix F.1.
> **3) Scaling GVN for Large-Scale Graphs**: There is no obvious solution for applying GVN to large-scale graphs. To address this, we proposed a partial fine-tuning strategy for GVN-Node and utilized a stored vector repository to save time and space costs.
> Therefore, the proposed methods are not trivial as they address unexplored but necessary challenges and thus possess technical novelties.
>
> 3. Overall SOTA Performance: We validated the effectiveness of GVN through extensive experiments conducted on seven widely-used benchmark datasets, including four challenging large-scale graph datasets (for instance, citation2, which contains 30,561,187 edges). Those experiments, performed across various metrics (as shown in Table 1 and Table 8 in Appendix D), demonstrate that the two variants of GVN can enhance the baseline GCN by 28.20% and 36.15% in terms of the relative performance. Moreover, GVNs are compatible with existing methods that utilize HSF (e.g., NCNC) and provide additional performance improvements that surpass the current state-of-the-art.
>
> In summary, GVN is a simple yet effective framework, which contrustively demonstrates vision benefit GNNs for the first time, and provide non-trivial solutions for unexplored but significant challenges in this path. Therefore, GVN has technical novelties.

---

> ### Author Response · Authors · 2024-11-20
> **Rebuttal to W2 and Q1 (Part 1)**
>
> > W2: The method is not convincing. It first transforms the graph data to visual modality and extracts the visual features as the structural features, which is expected to be more effective than the HSFs learned based on structural priors in graph modality. Intuitively, the visual features are more capable of learning appearance features or semantic features. In contrast, the graph data should have more expressive power than the visual modality in terms of the structural features.
>
> > Q1: It is suggested to given more theoretical analysis and qualitative comparison to show why the method can learn more effective structure features from the visual modality than the graph modality.
>
> We appreciate the feedback and would like to clarify our claims regarding Visual Structural Features (VSFs).
>
> **1) Complementary Role of VSFs:**
> We do not claim that VSFs are universally superior to all Heuristic Structural Features (HSFs). Instead, our results indicate that VSFs provide significant additional benefits when combined with HSFs. This synergy is evident as our model achieves state-of-the-art performance by leveraging both VSFs and advanced HSFs within the NCNC model, highlighting the complementary nature of VSFs.
>
> **2) Richness and Adaptability of VSFs:**
> VSFs incorporate a diverse range of structural heuristics, including common-neighbor-based methods (CN, AA, RA) and path-based methods (SPD), as detailed in Figure 2, Section 5.2. The table below shows the average accuracy of using VSFs (link-centered) to reproduce CN, AA, RA, and SPD HSFs, consistently exceeding 70%. This demonstrates that VSFs effectively function as a composite containing structural biases from various heuristics.
>
> |Average Reproduction Accuracy|Cora|Citeseer|Pubmed|
> |---|---|---|---|
> |VSF (before training)|75.43%|77.52%|72.31%|
> |VSF (after training)|73.20%|80.01%|75.35%|
>
> The richness of structural information in VSFs allows them to be multifunctional across different scenarios. Their learning capability enables them to adaptively adjust the emphasis on these diverse structural biases. The following table presents the performance of various structural features across different datasets (Hits@100 in Cora, Citeseer, Pubmed). It illustrates that: 1) Individual HSFs vary in effectiveness across datasets; for example, common-neighbor HSFs (CN, RA, AA) outperform path-based SPD in Cora, while the opposite is true in Citeseer and Pubmed. 2) As a rich source of structural information, VSFs consistently outperform individual HSFs. 3) The performance of VSFs is further enhanced through dataset-specific training.
>
> |Structural Features|Cora|Citeseer|Pubmed|
> |---|---|---|---|
> |CN|33.92±0.46|33.92±0.46|23.13±0.15|
> |AA|39.85±1.34|39.85±1.34|27.38±0.11|
> |RA|41.07±0.48|33.56±0.17|27.03±0.35|
> |SPD|29.97±0.76|54.24±0.21|51.70±0.25|
> |VSF (before training)|65.66±0.35|59.27±0.61|58.25±0.88|
> |VSF (after training)|**71.32±0.56**|**61.66±0.47**|**60.81±1.02**|
>
> Here we also want to demonstrate the adaptability of VSFs by analyzing their shifting preference for SPD-related information. In Cora, where SPD is less effective than common-neighbor HSFs, VSFs wisely reduce the SPD information (reproduction accuracy decreases from 73.38% to 51.33%) through adaptation to Cora-specific training. Conversely, this trend is reversed in Citeseer (increasing from 72.26% to 75.92%) and Pubmed (increasing from 70.82% to 76.14%), where SPD proves more effective than common-neighbor HSFs.
>
> Therefore, VSFs offer a **multifunctional and adaptable** approach to capturing structural features, complementing traditional HSFs, and enhancing overall model performance. By combining VSFs and HSFs, MPNN becomes more powerful in structural awareness.

---

> > ### Author Response · Authors · 2024-11-20
> > **Rebuttal to W2 and Q1(continue)**
> >
> > In addition, we demonstrated **how vision indeed aids GNNs in link prediction**. We acknowledge your point that **vision excels at capturing "appearance" features**. Our work, as detailed in the paper, shows that vision enhances link prediction by addressing two major limitations of Message Passing Neural Networks (MPNNs) in this context: 1) distinguishing isomorphic nodes, and 2) capturing local substructures. Those two issues are commonly noticed in previous link prediction works [r1-r4].
> >
> > 1. **Distinguishing Isomorphic Nodes**: As illustrated in Appendix A, vision can intuitively perceive the number of common neighbors or estimate path lengths. This capability directly assists in distinguishing isomorphic nodes, providing an advantage over traditional MPNNs.
> > 2. **Capturing Local Substructure**: As evidenced by the experiments in [r5], vision is adept at identifying local substructures such as cycles and connectivity. This ability complements the structural limitations of MPNNs.
> >
> > For further validation, we would like to redirect you to the experiments and analysis in Section 5.2. Those results demonstrate that VSFs significantly enhance MPNNs' ability to distinguish isomorphic nodes and capture local substructures (e.g., triangles) more effectively than MPNNs alone.
> >
> >
> > [r1] Can graph neural networks count substructures? NeurIPS 2020.
> >
> > [r2] Labeling Trick: A Theory of Using Graph Neural Networks for Multi-Node Representation Learning, NeurIPS 2021
> >
> > [r3] Graph neural networks for link prediction with subgraph sketching, ICLR 2023.
> >
> > [r4] Neural common neighbor with completion for link prediction, ICLR 2024.
> >
> > [r5] GITA: Graph to Image-Text Integration for Vision-Language Graph Reasoning, NeurIPS 2024.
> >
> > We sincerely appreciate the time you have dedicated to reviewing our manuscript and hope you can kindly consider our rebuttal.

---

> > > ### Author Response · Authors · 2024-11-24
> > > **Gentle Reminder: Looking Forward to Your Feedback on Rebuttal**
> > >
> > > Dear Reviewer G7xf,
> > >
> > > We greatly appreciate the time and effort you have dedicated to reviewing our work. As the deadline approaches, we want to ensure that our responses align with your expectations and address your concerns effectively.
> > >
> > > We recently submitted detailed responses to your comments and hope they have clarified the points you raised. If there are any areas where further clarification is needed or if you have additional insights, please feel free to reach out to us. We are eager to collaborate with you to enhance the manuscript and meet higher standards.
> > >
> > > Thank you once again for your time and thoughtful consideration.
> > >
> > > Best regards,
> > >
> > > The Authors

---

> > > > ### Author Response · Authors · 2024-11-25
> > > > **Gentle Reminder for Rebuttal Feedback**
> > > >
> > > > Dear Reviewer G7xf,
> > > >
> > > > We hope this message finds you well. We wanted to gently follow up regarding the feedback on our rebuttal submitted in response to your valuable comments. Understanding your insights is crucial for us. If there are any further clarifications needed or additional thoughts you wish to share, please feel free to discuss with us.  Thank you once again for your time and support.
> > > >
> > > > Warm regards,
> > > >
> > > > The Authors

---

> ### Comment · Reviewer_G7xf · 2024-11-29
>
> Thanks for the response, which partially convince me. My major concern still lies in the limited technical novelty. After comprehensive consideration, I raised my rating from 3 to 5.

---

> ### Author Response · Authors · 2024-11-30
> **Invitation to Discuss Key Contributions**
>
> Dear Reviewer G7xf,
>
> We sincerely appreciate your insightful and comprehensive review. We would like to offer further clarification regarding the contributions of our work in relation to your remaining concerns.
>
> Our research focuses on leveraging visual structural features (VSF) from visualized graph structures to enhance graph neural networks (GNNs). This approach, which integrates visual perception, is largely overlooked and novel within the graph learning community. Through empirical validation, we identified three key reasons why visual cues enhance GNN link prediction performance (Section 5.2). Our results demonstrate that incorporating VSF into the MPNN-based GVN framework significantly boosts performance while maintaining compatibility with existing methods (GVN improves both traditional MPNN GCN and advanced HF-enhanced MPNN NCCN with VSFs).
>
> We have made deliberate choices to ensure our approach is practical and compatible with existing solutions, which may have contributed to perceptions of simplicity. In Section 5.4, we provide the ablation and sensitivity analysis across five detailed aspects that may affect performance. To maintain the simplicity of the architecture, we also consciously chose to ablate techniques that are complex but empirically only marginally improve performance, and we report their empirical effects in Appendix F, H, I, J, and K.
>
> Therefore, we respectfully request you to consider whether exploring effective methods in a promising yet underexplored area should be undervalued due to their simplicity, especially when such simplicity arises from the pruning of incremental techniques and can lead to better compatibility with current mature technologies. We believe that simple yet impactful techniques can also drive meaningful advancements in the field.
>
> We sincerely thank you once again for your patience and kind consideration. We look forward to further discussing our work with you and hope to address any additional questions you may have.
>
> With deepest respect and gratitude,
>
> Authors

---

### Author Response · Authors · 2024-11-25
**Thank You for Your Review and Invitation for Further Discussion**

Dear Reviewers,

We hope this message finds you well. We would like to express our sincere gratitude for the thoughtful and detailed feedback you have provided on our manuscript. Your insights have been invaluable in guiding our revisions.

In response to your concerns, we have provided detailed explanations and conducted additional experiments to address each point raised. These responses and the corresponding results have been included in the revised manuscript. We have attached the updated version for your review.

We understand that your time is valuable, and we greatly appreciate your efforts in reviewing our rebuttal. As most reviewers have not yet had the opportunity to respond, we kindly ask for your feedback on whether the revisions meet your expectations. Your insights are crucial for us to ensure that all concerns have been adequately addressed.

Thank you once again for your dedication and support. We look forward to your valuable feedback and hope that our efforts have effectively resolved the issues you highlighted.

Best regards,

The Authors

---

### Meta-Review · Area_Chair_9XDJ · 2024-12-16

**Metareview:**

The paper proposes a message passing neural network that incorporates vision awareness by transforming the graph into images and extracting visual features from them. The objective is to provide fine-grained structural features to the model.

Strengths:
- The idea is simple.
- Integrating vision modality for link prediction is interesting.
- The experiments show effectiveness.

Weaknesses:
- Limited technical novelty. While the idea of using visual features is interesting, the techniques used are common.
- The motivation behind the use of visual features for graphs is unclear and unconvincing.
- The baselines should be reviewed and updated to include more recent methods.

There are major concerns regarding the limited novelty of the proposal and its insufficient motivation. The original concerns highlight the lack of baselines and problems with the experiments. Given these issues, I recommend rejecting the paper.

**Additional Comments On Reviewer Discussion:**

Reviewer G7xf was concerned about the limited technical contributions of the paper. Moreover, the reviewer commented that the method is not convincing since it transforms the graph into images to extract visual information. The reviewer argued that the graph should hold more information than the created image. The authors provided the reviewer with extensive descriptions addressing their concerns. However, the reviewer mentioned that the technical contributions were still limited and, while the reviewer increased the score from 3 to 5, this still reflects a negative evaluation of the paper.

Reviewer ZSnM found the idea of mixing a vision modality with MPNNs for link prediction interesting. However, the reviewer found the motivation lacking and in need of a more detailed description, as well as requiring theoretical motivation. The reviewer raised concerns about the compared methods, noting that they are not recent enough. The authors replied to the reviewer's concerns and presented some selected methods as comparisons. The reviewer, however, kept the score unchanged despite the comments.

Reviewer L8Vi mentioned that while the idea of using visual modalities is interesting, the techniques used are simple. Moreover, it wasn’t clear how the graph visualizer creates images that distinguish between graphs. The reviewer also raised concerns about the paper's presentation and clarity in the descriptions. The authors replied to the reviewer's concerns and explained the missing information. The reviewer did not respond to the comments.

Reviewer e7xg mentioned that the proposal's innovation is neutral. Moreover, the limited visual features used limit the structures that are extracted from the graph. Like other reviewers, this reviewer also raised concerns regarding the presentation of the paper. The authors presented additional results and discussions to address the reviewer’s comments. However, the reviewer raised additional questions about how the reconstruction of the original graph structure doesn’t seem to improve the performance of the proposal. In summary, the reviewer mentioned that the experiments do not validate the idea. After further discussion with the authors, the reviewer stated that the questions were addressed, yet maintained a weak positive score.

During the post-rebuttal discussion, I asked the reviewers about the lack of technical contributions and concerns regarding the lack of validation of the claims through the experiments. However, none of the reviewers replied.

Despite the authors' responses, I concur with the reviewers that the proposal lacks a clear description of the link between the visual features and the predictive performance, and that the experiments do not fully validate the claims. Despite the exchanges with some of the reviewers, the overall concerns of the paper outweigh the strengths mentioned. Therefore, I recommend the rejection of the paper.

---

### Decision · Program_Chairs · 2025-01-22

Reject